# Phosphorylation of luminal region of the SUN-domain protein Mps3 promotes nuclear envelope localization during meiosis

Hanumanthu BD Prasada Rao[1†], Takeshi Sato[2], Kiran Challa[1‡], Yurika Fujita[1], Miki Shinohara[1§], Akira Shinohara[1]*

[1]Institute for Protein Research, Osaka University, Suita, Japan; [2]Kyoto Pharmaceutical University, Kyoto, Japan

**Abstract** During meiosis, protein ensembles in the nuclear envelope (NE) containing SUN- and KASH-domain proteins, called linker nucleocytoskeleton and cytoskeleton (LINC) complex, promote the chromosome motion. Yeast SUN-domain protein, Mps3, forms multiple meiosis-specific ensembles on NE, which show dynamic localisation for chromosome motion; however, the mechanism by which these Mps3 ensembles are formed during meiosis remains largely unknown. Here, we showed that the cyclin-dependent protein kinase (CDK) and Dbf4-dependent Cdc7 protein kinase (DDK) regulate meiosis-specific dynamics of Mps3 on NE, particularly by mediating the resolution of Mps3 clusters and telomere clustering. We also found that the luminal region of Mps3 juxtaposed to the inner nuclear membrane is required for meiosis-specific localisation of Mps3 on NE. Negative charges introduced by meiosis-specific phosphorylation in the luminal region of Mps3 alter its interaction with negatively charged lipids by electric repulsion in reconstituted liposomes. Phospho-mimetic substitution in the luminal region suppresses the localisation of Mps3 via the inactivation of CDK or DDK. Our study revealed multi-layered phosphorylation-dependent regulation of the localisation of Mps3 on NE for meiotic chromosome motion and NE remodelling.

*For correspondence:
ashino@protein.osaka-u.ac.jp

Present address: [†]National Institute for Animal Biotechnology, Hyderabad, India; [‡]Paul Scherrer Institute, Villigen, Switzerland; [§]Graduate School of Agriculture, Kindai University, Nara, Japan

Competing interest: The authors declare that no competing interests exist.

## Introduction

During meiosis, homologous chromosomes pair with each other. This pairing leads to juxtaposition of the chromosomes along their entire length, resulting in the formation of the synaptonemal complex (SC). SC is a proteinaceous structure comprising two chromosome axes called the axial/lateral element (AE/LE), flanking a central region with a ladder-like structure (*Hawley and Gilliland, 2009*; *Zickler and Kleckner, 1999*). AE formation is initiated in the leptotene stage of meiotic prophase I, accompanied by the assembly of the chromosome axis with multiple chromatin loops. In the zygotene stage, some homologous AE pairs form a short SC, while in the pachytene stage, the SC elongates along the entire chromosome. After the pachytene stage, the SC dismantles in the diplotene stage before the onset of meiosis I.

Meiotic cells also exhibit a unique arrangement of chromosomes in the nucleus. In most organisms, telomeres attach to nuclear envelopes (NEs) in early meiotic prophase (or premeiotic phase) and occasionally cluster in one area of NE, particularly near the vicinity of the centrosome (a microtubule organising centre) (*Zickler and Kleckner, 1998*). Telomere clustering is seen only in the zygotene stage and is known as 'telomere/chromosome bouquet'. In the pachytene stage, telomere bouquets are resolved, and telomeres are dispersed over the entire NE. The bouquet configuration of chromosomes

is proposed to promote homologous pairing by restricting the homology search process from three dimensions to two dimensions (*Zickler and Kleckner, 1998*).

The dynamic nature of meiotic chromosomes is conserved between yeasts and mammals (*Hiraoka and Dernburg, 2009*; *Koszul and Kleckner, 2009*; *Burke, 2018*; *Lee et al., 2020b*; *Paouneskou and Jantsch, 2019*). The oscillating motion of nuclei during meiotic prophase I has been well described in fission yeast (*Chikashige et al., 1994*). Nuclear movement is driven by cytoplasmic microtubule cables, along which the spindle pole body (SPB), an yeast centrosome equivalent, embedded in NE moves together with the clustered telomeres. This telomere-led movement promotes the pairing of homologous loci on chromosomes. Although less coordinated, chromosome motion throughout prophase I has been reported in budding yeast, nematodes, maize, and mouse (*Lee et al., 2015*; *Penkner et al., 2009*; *Sheehan and Pawlowski, 2009*; *Shibuya et al., 2014*).

Protein ensembles embedded in NE that connect telomeres in the nucleoplasm with the cytoskeleton in the cytoplasm, which are referred to as the linker of nucleocytoskeleton and cytoskeleton (LINC) complex, promote meiotic chromosome motion (*Hiraoka and Dernburg, 2009*; *Starr and Fridolfsson, 2010*; *Jahed et al., 2021*; *Lee et al., 2020b*; *Wong et al., 2021*). The LINC complex necessary for chromosome motion contains an inner nuclear membrane (INM) protein harbouring the SUN (Sad1/UNc-84) domain and an outer nuclear membrane (ONM) protein with the KASH (Klarsicht, ANC-1, SYNE1 Homology) domain. The SUN and KASH domains interact with each other in a space between the INM and ONM, luminal region, or peri-nuclear space. SUN-domain proteins bind to telomeres and/or nucleocytoskeletons in the nucleoplasm, while KASH-domain proteins bind to cytoskeletons. LINC complexes transmit forces generated by the cytoskeleton to NEs for nuclear positioning and migration in somatic cells and telomeres for chromosome motion in meiotic cells (*Starr and Fridolfsson, 2010*).

In *Saccharomyces cerevisiae*, telomeres are tethered to NEs as several clusters during mitosis and move around NEs during meiotic prophase I. These telomeres are transiently clustered during prophase I (*Conrad et al., 2008*; *Koszul et al., 2008*; *Trelles-Sticken et al., 2000*; *Trelles-Sticken et al., 1999*). Telomere movement on NE drives chromosome motion inside the nucleus (*Koszul et al., 2008*; *Scherthan et al., 2007*). At least three distinct regulatory steps facilitate chromosome motion during budding yeast meiosis. First, meiosis-specific components of the LINC complex, such as Ndj1 and Csm4, are expressed and incorporated. Ndj1 is a meiosis-specific telomere-binding protein necessary for telomere tethering to NEs (*Chua and Roeder, 1997*; *Conrad et al., 1997*), whereas Csm4 is a meiosis-specific ONM protein (*Conrad et al., 2007*; *Kosaka et al., 2008*; *Wanat et al., 2008*), which interacts with a unique class of KASH protein, Mps2 (*Chen et al., 2019*; *Lee et al., 2020a*). Second, actin cable formation is specifically induced in the cytoplasm of meiotic cells (*Koszul et al., 2008*; *Taxis et al., 2006*). Third, NE dynamics, including NE remodelling and movement, occur prominently in meiotic prophase I (*Conrad et al., 2007*; this paper). NE remodelling is accompanied by NE localisation of a yeast SUN-domain protein, Mps3/Nep98 (*Conrad et al., 2007*), and Mps2 (*Lee et al., 2020a*). During mitosis, Mps3 and Mps2 predominantly localise to SPB as a component of the half-bridge as well as SPB-interacting network (SPIN) (*Chen et al., 2019*) and play an essential role in SPB duplication (*Jaspersen et al., 2002*; *Nishikawa et al., 2003*). During meiotic prophase-I, in addition to SPB, Mps3 localises as a distinct protein ensemble on NE as a LINC complex (*Conrad et al., 2007*). In early meiotic prophase I, Mps3 forms several distinct foci/patches on NE and later shows more dispersed staining around the NE. Mps3 plays an important role in the telomere/chromosome dynamics (*Conrad et al., 2007*). However, how meiosis-specific Mps3 localisation in NE is induced is poorly understood.

Here, we showed that cyclin-dependent Cdc28 kinase (CDK) and Dbf4-dependent Cdc7 kinase (DDK) are necessary for dynamic telomere movement, particularly for the resolution of telomere clusters. Defects in telomere movement induced by CDK and DDK inactivation are associated with impaired NE localisation of Mps3 and resolution of Mps3 clusters. We also found that the luminal region of Mps3, located near the INM, is important for NE localisation. This region is subject to phosphorylation during prophase I in meiosis. Phospho-defective *mps3* mutant protein showed reduced NE localisation during meiosis. The introduction of negatively charged residues in this region suppresses the defects in the Mps3 resolution in *cdk* and *ddk* mutants. In reconstituted liposomes, an Mps3 peptide containing the luminal region with negatively charged residues showed reduced affinity to the lipid membrane relative to the control wild-type Mps3 peptide. Our results suggest that multiple

phosphorylation events, such as CDK and DDK-dependent phosphorylation as well as phosphorylation in the luminal region, control the Mps3 localisation to NE, thereby also controlling the assembly of meiosis-specific canonical LINC complex for chromosome motion and NE remodelling during meiosis.

## Results

### Force-independent localization of Mps3 on NE during meiosis

We examined the localisation of the Mps3 fusion protein with a green fluorescent protein (GFP), Mps3-GFP (*Conrad et al., 2007*), in various mutants. Mps3-GFP showed stage-specific distribution and movement in NE during meiotic prophase I (*Figure 1A and B*). At 0 h, Mps3 predominantly localised as a single focus, probably on SPB (*Figure 1A*; *Jaspersen et al., 2002*; *Nishikawa et al., 2003*) with little movement (*Figure 1C*; *Figure 1—video 1*). During early prophase I, including meiotic S phase (2–3 h), Mps3 formed 2–5 foci (including one focus in SPB) on NE with slow movement. After 3 h, the number of Mps3 foci increased to more than 5, and several Mps3 foci fused to form a patch (*Figure 1A–C*; *Figure 1—videos 2–4*), as shown previously (*Conrad et al., 2007*). During this stage, Mps3 foci/patches moved around the NE with oscillatory motion (*Figure 1—figure supplement 1*). At 3–5 h, some Mps3 foci/patches were localised to one area of NE with clustering. In late prophase I (e.g. at 5 and 6 h), Mps3 often covered most of the NE and was associated with NE deformation and protrusion (*Figure 1C*; *Figure 1—videos 3 and 4*). Mps3 localisation was classified into four classes and quantified at each time point (*Figure 1B*). After 7 h, most Mps3 signals on NE disappeared, leaving two SPB-associated foci (*Figure 1A*).

Although Mps3 movement is dependent on Csm4 (*Conrad et al., 2007*; *Kosaka et al., 2008*; *Wanat et al., 2008*) and partly on Ndj1 (*Chua and Roeder, 1997*; *Conrad et al., 1997*), Mps3 localisation on NE did not depend on Ndj1 and Csm4 (*Figure 1D*). At late times, such as 5 h, the *ndj1* mutant showed full coverage of Mps3 on NE (*Conrad et al., 2007*) with NE deformation (distorted nucleus) and motion. The *csm4* mutant formed multiple Mps3 patches/foci on NE (*Conrad et al., 2007*; *Kosaka et al., 2008*; *Wanat et al., 2008*), but did not show full coverage of NE. The *csm4* mutant is also defective in clustering and motion of Mps3 foci/patches, as well as NE deformation with a round nucleus (*Figure 1D*). Moreover, we examined the effect of an actin polymerisation inhibitor, latrunculin B (LatB), on Mps3 localization (*Trelles-Sticken et al., 2005*). Treatment of meiotic cells with LatB, which disrupted the cytoplasmic actin cables but not the actin patches (*Figure 1—figure supplement 1*), showed normal distribution of Mps3 foci/patches on NE during prophase (*Figure 1D*). However, the dynamic movement of Mps3-GFP was largely absent (*Figure 1—figure supplement 1*; *Figure 1—video 5*) without the deformation of nuclei. These defects were similar to those observed in the *csm4* mutant (*Figure 1—figure supplement 1*). One hour after washing out LatB, the motion of Mps3 resumed (*Figure 1D*). These results suggest that Mps3 dynamics during meiosis can be functionally separated into two processes: force-independent NE localisation and force-dependent motion. Force-dependent motion requires actin polymerisation, Csm4, and a partly, Ndj1. On the other hand, force-independent NE localisation does not require actin polymerisation or Csm4 (Ndj1), suggesting a unique localisation mechanism of Mps3 on NE during meiosis.

### CDK and DDK promote proper Mps3 localization on NE during meiosis

Because Mps3-dependent telomere movement is initiated in early meiotic prophase (*Conrad et al., 2008*; *Conrad et al., 2007*), we analysed the role of two protein kinases that are active during early meiosis: CDK and DDK. Because CDK and DDK are essential for vegetative growth, we used conditional mutant alleles of the catalytic subunits of CDK and DDK (Cdc28 and Cdc7), *cdc28-as1*, and *cdc7-as3*, which are sensitive to specific inhibitors,1 NM-PP1 and PP1, respectively (*Benjamin et al., 2003*; *Wan et al., 2006*). Although CDK and DDK are also essential for DNA replication and recombination in meiosis (*Benjamin et al., 2003*; *Sasanuma et al., 2008*), telomere movement during meiosis and Mps3 localisation are independent of replication and recombination (*Conrad et al., 2007*; *Trelles-Sticken et al., 1999*).

We performed time-lapse analysis of Mps3-GFP in *cdc28-as1* and *cdc7-as3* in the absence or presence of each specific inhibitor (*Figure 1E*). To inactivate CDK, 1 NM-PP1, which did not affect Mps3-GFP dynamics in wild-type cells or a steady-state level of Mps3-Flag protein in *cdc28-as1* cells (*Figure 1—figure supplement 1*), was added to *cdc28-as1* at 0 h. In the absence of the inhibitor, the

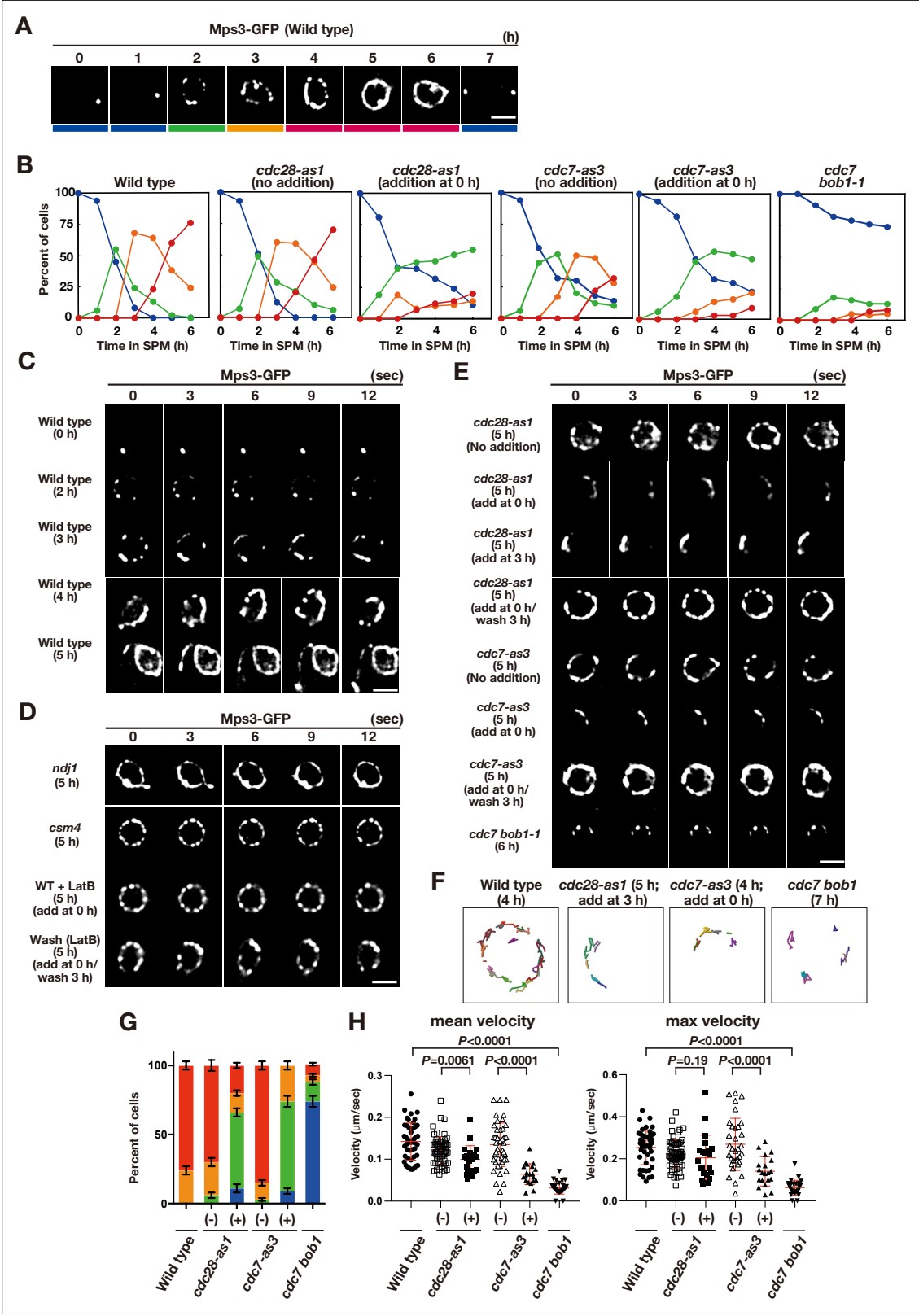

**Figure 1.** Meiotic Mps3 localization and movement on NEs. (**A**) Mps3-GFP localization was analyzed in wild-type cells (PRY64) at various time points during meiosis. Representative images at each time are shown. Bottom color bars indicate classes of Mps3-localization in (**B**). Bar indicates 2 μm. (**B**) Based on Mps3-GFP localization, cells with Mps3-GFP were classified into four classes (**A**) and quantified: single Mps3 focus (blue), 2–5 foci (green), more than five foci/patches (orange), and coverage of the Mps3 signal on NE (red). At each time point, more than 100 nuclei were counted. The graphs

*Figure 1 continued on next page*

*Figure 1 continued*

are a representative of two independent time courses. (**C**) Time-lapse analysis of Mps3-GFP in the wild-type strain (PRY64) at different time points in meiosis. A single focal plane of a cell every 3 s is shown. See *Figure 1—videos 1–4*. (**D**) Time-lapse analysis of Mps3-GFP in various strains at different time points in meiosis. A single focal plane of a cell was analyzed every 3 s. The inhibitor LatB (30 µM) was added at 0 h. See *Figure 1—video 5*. While washing, the inhibitor was washed at 5 h, and the cells were analyzed at 6 h. Wild type, PRY64; *ndj1*, PRY192; *csm4*, PRY198. (**E**) Time-lapse analysis of Mps3-GFP in the *cdc28-as1* mutant (PRY71) treated with or without the inhibitor 1NM-PP1 (0.5 µM) at 5 h during meiosis. The inhibitor was added at 0 h (second panels) or 3 h (third panels). While washing (fourth panels), the inhibitor was added at 0 h and washed at 3 h, and the cells were analyzed at 5 h. Time-lapse analysis of Mps3-GFP in the *cdc7-as3* mutant (PRY260) treated with or without the inhibitor PP1 (15 µM). The *cdc7 bob1-1* mutant (PRY115) was analyzed at 6 h. See *Figure 1—videos 6–8*. (**F**) Tracking of Mps3-GFP in wild type, *cdc28-as1* (PRY71 with the inhibitor), *cdc7-as3* (PRY260 with the inhibitor), and *cdc7 bob1-1* (PRY115). Tracking was monitored for all Mps3 foci/patches in a single cell for 20 s at 4 or 5 h in SPM. Each line represents tracking of the foci at a single focal plane. (**G**) Percentages of cells with different classes of Mps3-GFP were quantified at 5 h under different conditions (triplicates, Error bars show standard deviation; s.d.); single Mps3 focus (blue), 2–5 foci (green), more than five foci/patches (orange), and coverage of the Mps3 signal on NE (red). (**H**) Velocity of Mps3-GFP foci or patches was quantified. Time-lapse images were taken for every second in 20 s. Mps3-GFP foci or patches were identified as shown in Materials and Methods and followed for their tracks. For each track, an average velocity and maximum velocity were calculated. More than 20 cells were analyzed for the quantification. Red lines show mean with s.d. p-Values were calculated using Mann-Whitney's *U*-test.

The online version of this article includes the following video, source data, and figure supplement(s) for figure 1:

**Source data 1.** Source data for *Figure 1*.

**Figure supplement 1.** Meiotic movements of Mps3 and Rap1 in the presence of the inhibitors.

**Figure 1—video 1.** Mps3-GFP in a wild-type cell at 0 h in SPM.
https://elifesciences.org/articles/63119/figures#fig1video1

**Figure 1—video 2.** Mps3-GFP in a wild-type cell at 4 h in SPM.
https://elifesciences.org/articles/63119/figures#fig1video2

**Figure 1—video 3.** Mps3-GFP in a wild-type cell at 5 h in SPM, showing deformation.
https://elifesciences.org/articles/63119/figures#fig1video3

**Figure 1—video 4.** Mps3-GFP in a wild-type cell at 5 h in SPM, showing protrusion.
https://elifesciences.org/articles/63119/figures#fig1video4

**Figure 1—video 5.** Mps3-GFP in a wild-type cell treated with LatB (0 h) at 5 h in SPM.
https://elifesciences.org/articles/63119/figures#fig1video5

**Figure 1—video 6.** Mps3-GFP in *cdc28-as1* cells treated with 1NM-PP1 (0 h) at 5 h in SPM.
https://elifesciences.org/articles/63119/figures#fig1video6

**Figure 1—video 7.** Mps3-GFP in a *cdc7-as3* cell treated with PP1 (0 h) at 5 h in SPM.
https://elifesciences.org/articles/63119/figures#fig1video7

**Figure 1—video 8.** Mps3-GFP in a *cdc7 bob1-1* cell at 5 h in SPM.
https://elifesciences.org/articles/63119/figures#fig1video8

mutant cells showed normal Mps3 localisation during meiosis (*Figure 1E*). CDK inactivation impaired Mps3 localisation and motion (*Figure 1B and E*). In the wild type, several Mps3 foci/patches are loosely clustered on one area of NE in a transient manner, which reflects telomere bouquet (*Conrad et al., 2007*). With CDK inhibition, these loose clusters of Mps3 foci were persistent during prophase I, which largely reduced the spread of Mps3 foci on NE (*Figure 1B and G*). Tracking the motion of Mps3 foci confirmed persistent clustering with restricted motion of the foci in one region of NE under these conditions (*Figure 1F*).

We also measured the speed of Mps3 GFP focus/patches on NE (*Figure 1H*). Previous measurement of the motion of a single chromosome locus showed heterologous nature of chromosome motion (*Conrad et al., 2008*). To avoid biased results, we simply measured a step-size of a single Mps3-GFP focus or patch in one-second interval through 20 s in multiple nuclei, and measured an average velocity a maximum velocity during the measurement (*Figure 1H*). Wild type Mps3-GFP at 5 h showed 0.19 and 0.25 µm/s in average and maximum speeds, respectively. CDK inactivation slightly decreased the average speed, but not the maximum speed (*Figure 1H*, *Figure 1—video 6*).

Importantly, 2 h after washing out the inhibitor, NE localisation and Mps3 motion were recovered. CDK inactivation after meiotic S phase (the addition of the inhibitor at 3 h) interfered with the dispersed localisation of Mps3 throughout NE with loose clustering of Mps3 foci/patches (*Figure 1E*). Therefore, persistent CDK activity is necessary for the establishment and maintenance of the dispersed

localisation of Mps3, and thereby the resolution of clustering. We concluded that CDK plays a role in the resolution of Mps3 focus clustering (probably, not in focus/patch formation per se) and a minor role in the motion of Mps3 on NE.

DDK inactivation (*cdc7-as3*+ PP1) also induced a defect in Mps3 localisation on NE and motion (*Figure 1B and E*). PP1 did not affect the localisation of Mps3-GFP in wild-type cells or a steady level of Mps3-Flag protein under DDK inactivation conditions (*Figure 1—figure supplement 1*). In its absence, *cdc7-as3* cells showed normal localisation and motion of Mps3 foci/patches (*Figure 1B and E*). DDK inhibition by PP1 from 0 h reduced the NE localisation of Mps3 with accumulation of the loose cluster of the foci (*Figure 1E and F*). Removal of the inhibitor restored the NE localisation of Mps3 in *cdc7-as3* cells (*Figure 1E*). The addition of PP1 at 3 h also induced a Mps3 localisation defect similar to that at 0 h. This shows that persistent DDK activity is critical for the efficient resolution of Mps3 clusters on NE. The resolution defect of Mps3 clusters in reduced DDK activity is similar to that in response to decreased CDK activity, suggesting that CDK and DDK work in the same pathway to control Mps3 dynamics during meiosis. In contrast to reduced CDK, reduced DDK activity clearly decreased the velocity of Mps3 foci on NE (*Figure 1H*, *Figure 1—video 7*). Therefore, DDK appears to play a more critical role in Mps3 motion than CDK.

We also analysed the Mps3-GFP localisation in the *cdc7 bob-1–1* double mutant, in which the *bob1-1* mutation in the helicase gene *MCM5* suppresses the lethality of the *cdc7* deletion (*Jackson et al., 1993*; *Sasanuma et al., 2008*), A small fraction (~20%) of the *cdc7 bob-1* mutant cells showed a few Mps3 foci on NE, whereas most of the cells did not form any Mps3 foci on NE (*Figure 1B, E and G*). The velocity of Mps3 foci formed in the complete absence of DDK was highly reduced and was similar to that in the *csm4* mutant and in the presence of LatB (*Figure 1—figure supplement 1*; *Figure 1—video 8*). These defects in the *cdc7 bob-1–1* double mutant were more severe than those in the *cdc7-as3* mutant with PP1, confirming the importance of DDK in Mps3 localisation and motion in NE.

Whole cell immunostaining of the Mps3-Flag and Ndj1-HA proteins showed that reduced CDK and DDK activities hampered the localisation of Mps3, but not of Ndj1, to NE during meiosis (*Figure 1—figure supplement 1*), suggesting a critical role for the kinases in the coupling of Mps3 to Ndj1 on NE. Double-staining of Mps3 and Ndj1 in wild-type meiosis showed distinct Mps3- and Ndj1-staining domains on NE together with colocalized foci/patches of both. The reduced colocalization of Ndj1 and Mps3 on NE is clearly seen under CDK and DDK inactivation conditions. Ndj1 was not required for Mps3 localisation on the NE (*Figure 1D*). These results suggest the presence of an Mps3-independent pathway for Ndj1 localisation on NE or telomeres, probably mediated by Ndj1 binding to Rap1/telomeres. Taken together, these results revealed that CDK and DDK activities are necessary for proper Mps3 localisation and efficient motion in NE during meiosis.

## CDK and DDK promote telomere dynamics during meiosis

To determine the role of CDK and DDK in telomere movement, we analysed the motion of GFP fusion of a telomere-binding protein, Rap1, as described previously (*Trelles-Sticken et al., 2005*). During vegetative growth or pre-meiosis, Rap1-GFP exhibited 1–4 foci on NE on a single focal plane with little movement (*Figure 2A and B*; *Figure 2—video 1*; *Trelles-Sticken et al., 2005*). Once the cells entered meiosis, Rap1 foci on NE increased in number (up to 10). Rap1-GFP foci moved rapidly and were occasionally clustered in one area of NE (*Figure 2A and B*; *Figure 2—video 2*), which corresponds to the telomere bouquet (*Trelles-Sticken et al., 2005*). This telomere clustering was very transient (half-life less than 5 s) and was predominantly observed after 3–4 h of incubation in sporulation medium (*Figure 2A and D*). Rap1 foci were more dispersed during late meiotic prophase (4–6 h for the pachytene stage) before the onset of meiosis I (MI; *Figure 2D*). Tracking analysis showed that Rap1 foci move around the meiotic nucleus in an uncoordinated manner (*Figure 2B*).

CDK inactivation (*cdc28-as1* + 1 NM-PP1) impaired meiotic Rap1 dynamics (*Figure 2B and E*). With compromised CDK activity from 0 h, cells with clustered Rap1 foci accumulated during prophase I (*Figure 2D and E*; *Figure 2—video 3*), indicating that CDK activity is required for meiotic telomere dynamics, particularly for the resolution of telomere clusters (but not necessarily for clustering per se). The 2D tracking confirmed the restricted motion of Rap1 foci in one nuclear area (*Figure 2B*). The addition of the inhibitor during early prophase I (e.g. at 3 h) also resulted in defective resolution of telomere clustering; washing off the inhibitor restored telomere dynamics (*Figure 2E*). Treatment of

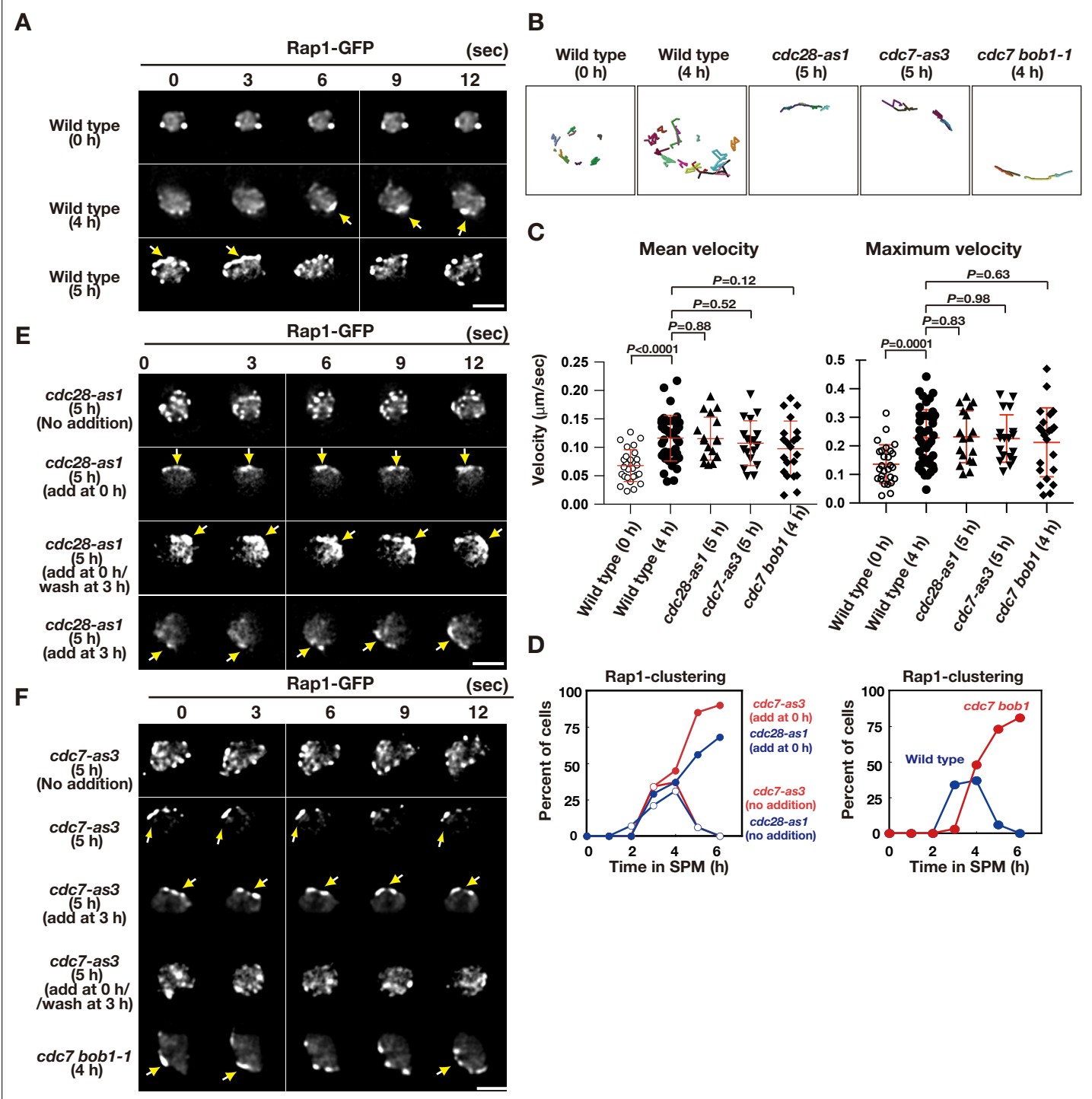

**Figure 2.** Meiotic telomere movement in *cdk* and *ddk* mutants. (**A**) Time-lapse analysis of Rap1-GFP in wild type (HKY167) at different time points during meiosis. An image of a single focal plane of each mutant cell was taken every 3 s. Clustering of telomeres is shown by yellow arrows. Bar indicates 2 µm. See *Figure 2—videos 1 and 2*. (**B**) Tracking of Rap1-GFP in wild type (0 and 4 h), *cdc28-as1* (with the inhibitor), and *cdc7-as3* (with the inhibitor). Tracking was monitored for all Rap1 foci in a single cell for 20 s. Each line represents tracking of the center of foci at a single focal plane. (**C**) Velocity of Rap1-GFP foci or patches were quantified. Time-lapse images were taken for every second in 20 s. Rap1-GFP foci or patches were identified as shown in Methods and followed for their tracks. For each track, an average velocity and maximum velocity were calculated. More than 20 cells were analyzed for the quantification. Red lines show mean with s.d. p-Values were calculated using Man-Whitney's *U*-test. (**D**) Kinetics of telomere clustering during meiosis were quantified. At each time point, more than 150 nuclei were counted for clustering of telomeres (arrows in **A, E, F**). The graphs are a representative of two independent time course. (left) *cdc28-as1* with 1NM-PP1, blue closed circles; *cdc28-as1* without 1NM-PP1, blue closed circles;

*Figure 2 continued on next page*

*Figure 2 continued*

*cdc7-as3* with PP1, red closed circles; *cdc7-as3* without PP1, red closed circles. (right) wild type, blue closed circles; *cdc7 bob1-1*, red closed circles. (**E**) Time-lapse analysis of Rap1-GFP in various strains (*cdc28-as1*, PRY68) at different time points during meiosis. An image of a single focal plane of each mutant cell was taken every 3 s. The inhibitors (final 0.5 μM of 1NM-PP1) were added at 0 h (second panels) or 3 h (third panels). While washing (bottom panels), the inhibitor was added at 0 h and washed at 3 h, and the cells were analyzed at 5 h. Clustering of telomeres is shown by yellow arrows. Bar indicates 2 μm. See *Figure 2—video 3*. (**F**) Time-lapse analysis of Rap1-GFP in various strains (*cdc7-as3*, PRY79; *cdc7 bob1-1*, PRY116) at different time points during meiosis. An image of a single focal plane of each mutant cell was taken every 3 s. For the *cdc7-as3* mutant, the inhibitors (final 15 μM PP1, respectively) were added at 0 h (second panels) or 3 h (third panels). While washing (bottom panels), the inhibitor was added at 0 h and washed at 3 h, and the cells were analyzed at 5 h. Clustering of telomeres is shown by yellow arrows. Bar indicates 2 μm. See *Figure 2—video 4* and *Figure 2—video 5*.

The online version of this article includes the following video, source data, and figure supplement(s) for figure 2:

**Source data 1.** Source data for *Figure 2*.

**Figure supplement 1.** Meiotic movements of Mps3 in the presence of the inhibitors.

**Figure 2—video 1.** Rap1-GFP in a wild-type cell at 0 h in SPM.
https://elifesciences.org/articles/63119/figures#fig2video1

**Figure 2—video 2.** Rap1-GFP in a wild-type cells at 4 h in SPM.
https://elifesciences.org/articles/63119/figures#fig2video2

**Figure 2—video 3.** Rap1-GFP in a *cdc28-as1* cell treated with 1NM-PP1 (0 h) at 5 h in SPM.
https://elifesciences.org/articles/63119/figures#fig2video3

**Figure 2—video 4.** Rap1-GFP in a *cdc7-as3* cell treated with PP1 (0 h) at 5 h in SPM.
https://elifesciences.org/articles/63119/figures#fig2video4

**Figure 2—video 5.** Rap1-GFP in a *cdc7 bob1-1* cell at 5 h in SPM.
https://elifesciences.org/articles/63119/figures#fig2video5

wild-type cells with the inhibitor did not affect Rap1-GFP dynamics, confirming the specificity of the inhibitor (*Figure 2—figure supplement 1*). These results indicate that persistent CDK activity during meiotic prophase I promotes normal telomere dynamics. CDK inactivation does not affect the formation of cytoplasmic actin cables during meiosis (*Figure 1—figure supplement 1*).

Similar to CDK inactivation, Cdc7 inactivation (the treatment of *cdc7-as3* with PP1) at 0 and 3 h impaired Rap1-GFP dynamics, which showed accumulation of Rap1 clusters with restricted motion (*Figure 2B–D and F*; *Figure 2—video 4*); however, treatment of wild-type cells with PP1 did not affect Rap1 dynamics (*Figure 2—figure supplement 1*). This defective resolution of the Rap1 cluster under DDK inhibition was similar to that observed in CDK inhibition. Moreover, the *cdc7* null mutant with *bob1-1* mutation exhibited persistent telomere clustering (*Figure 2B, D and F*, and *Figure 2—figure supplement 1*). These results indicate that persistent Cdc7 kinase activity is also required for meiotic telomere dynamics. Although CDK and DDK inactivation restricted Rap1 foci to one area, inactivation did not affect the velocity of Rap1 foci (*Figure 2C*). Since Mps3 foci are actually formed under CDK and DDK inactivation conditions (*Figure 1*), the Mps3 complexes on NE are sufficient to ensure some Rap1 motion. Alternatively, the Mps3-independent mechanism may promote motion under reduced CDK or DDK activities, particularly in the *cdc7* null mutant.

## Mps3 is phosphorylated during meiosis

Previously, it was shown that S70 of SPB-associated Mps3 is phosphorylated in late prophase-I (*Li et al., 2017*; *Figure 3A*). We also found that Mps3 is a phosphoprotein in middle prophase I (*Figure 3*). On immunoprecipitation (IP) western blots, Mps3-Flag showed a mobility shift, which was induced during meiotic prophase I (4 and 6 h; *Figure 3B*, left panel). Mps3 mobility in the IP fractions was reduced by treatment with calf intenstine phosphatase (*Figure 3C*). Moreover, IP fractions of Mps3-Flag from meiotic cells showed that Mps3 cross-reacted with anti-phosphoserine antibody (*Figure 3D*), whose reactivity was largely diminished by treatment with phosphatase.

Previous systematic analyses of CDK substrates in vitro identified Mps3 as a Cdc28–Clb2 substrate (*Ubersax et al., 2003*). Mps3 contains six potential CDK sites (S/TPXK/R or S/TP) located in the luminal region, two of which seem to be candidates for CDK/DDK phosphorylation (188–190 TSSPGK and 450–451 TSP; *Figure 3A and E*). Because the 450–451 site is located in the SUN domain, which is not essential for Mps3 localisation on NE (*Rao et al., 2011*), we focused on T188-S189-S190 (hereafter,

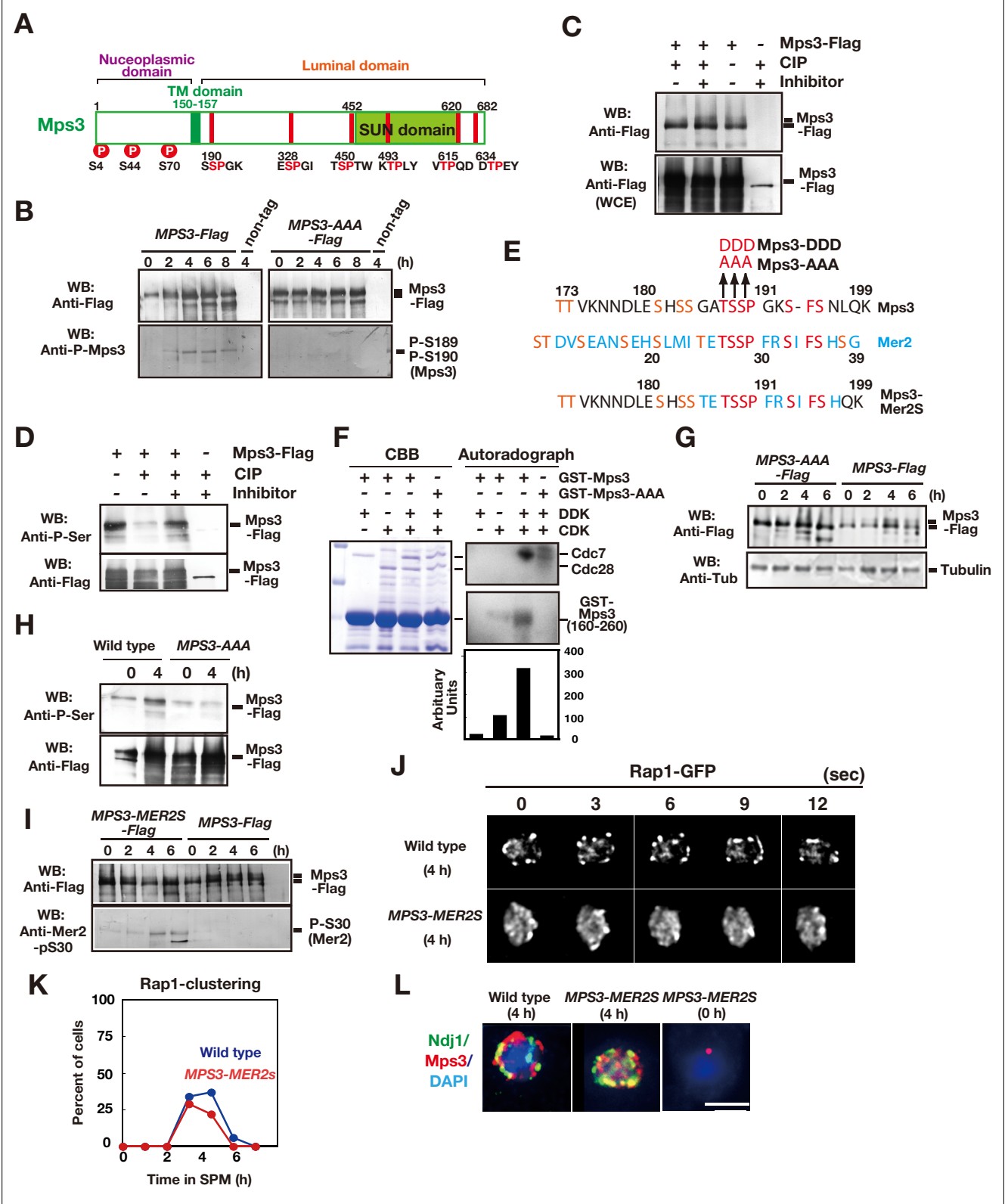

**Figure 3.** Mps3 is phosphorylated in luminal region. (**A**) Schematic representation of the Mps3 protein. The putative transmembrane (TM) region (dark green) and SUN domain (green) are shown with three known phosphorylation sites (red circles) in nucleoplasmic region (*Lanz et al., 2021*). The position of a CDK consensus site (S/TP) is shown in red lines with aa sequence on the bottom. All the six sites are located in a luminal region of Mps3. (**B**) Western blotting of the Mps3-Flag protein during meiosis. Wild-type (left, HKY404) and *mps3-AAA* mutant (middle, PRY163) proteinwere analyzed

*Figure 3 continued on next page*

*Figure 3 continued*

at different time points. Fractions immunoprecipitated with anti-Flag antibody were probed with anti-Flag antibody (top) as well as a phospho-specific antibody (bottom) that recognized T189 and S190 phosphorylation. (**C**) In vivo phosphorylation was confirmed by a decreased band shift of Mps3-Flag protein purified from meiotic cell lysates (4 h incubation with SPM) using the anti-Flag antibody. After immuno-precipitation, the precipitate was incubated with 1 unit of CIP for one hour either in the presence or absence of phosphatase inhibitors. Mps3-Flag was detected using anti-Flag (M2) antibody. (**D**) Affinity-purified-Mps3-Flag from meiotic cell lysates (4 h incubation with SPM) using anti-Flag column was treated with CIP as described in (**A**) and was probed with anti-Phospho-Serine antibody (Qiagen). During incubation, Mps3 protein was degraded extensively (bottom). (**E**) Sequence homology of an Mps3 region with a CDK/DDK phosphorylation site of the Mer2 protein. Identical amino acids are shown in red. Putative CDK or DDK phosphorylation sites are shown in orange. Unique amino acid sequence in Mer2 is shown in pale blue. The substitutions (*mps3-AAA*, *MPS3-DDD*, and *MPS3-MER2S*) for the putative phosphorylation sites of Mps3 are shown on the top. (**F**) In vitro phosphorylation analysis of GST-Mps3 fusion proteins. Purified wild-type GST-Mps3 and GST-Mps3-AAA fragments (shown on the left by Coomassie staining) were incubated with different combinations of partially purified the Cdc28–Clb5 complex and/or Cdc7–Dbf4 kinase complexes in the presence of $\gamma$ -$^{32}$P-ATP. After 1 h incubation, proteins were fractionated on a SDS-PAGE gel and analyzed using a phosphorimager for incorporation of $^{32}$P in the Mps3 fragments (right bottom) and Cdc28/Cdc7 protein (right top). Cdc7 was phosphorylated by CDK. Quantification is shown below the autoradiogram. (**G**) Western blotting of Mps3 and tubulin at various times in meiosis. Cell lysates from the *MPS3-Flag* (right, HKY404) and *MPS3-AAA-Flag* (left, PRY163) strains were analyzed by western blotting using anti-Flag and anti-tubulin antibodies. The wild-type Mps3 protein shows band shift during meiosis for example at 4 and 6 h. (**H**) Reactivity of wild-type Mps3-Flag and Mps3-AAA mutant protein to anti-phospho-serine antibody was examined by western blotting. wild type Mps3-Flag and Mps3-AAA mutant protein at each time point were IPed and probed with the antibody. While wild type Mps3-Flag shows an increase reactivity to the antibody, Mps3-AAA-Flag did not increase the reactivity in meiosis (4 h) compared to mitosis (0 h). (**I**) Western blotting of the Mps3-Flag protein during meiosis. Wild-type (left, HKY404) and *MPS3-MER2S* proteins (right, PRY514) were analyzed at different time points. Fractions immunoprecipitated with anti-Flag antibody were probed with anti-Flag and anti-Mer2-phospho-S30 antibodies. (**J**) Rap1-GFP dynamics at 4 h of meiosis analyzed in *MPS3-MER2s* (PRY518) and wild-type (HKY167) strains. (**K**) Kinetics of Rap1 clustering in each strain is shown in G. Counting was performed as described in *Figure 2D*. Blue, wild type; red, *MPS3-MER2S*. (**L**) Localization of Ndj1-HA (green), Mps3-Mer2S-Flag (red), and DAPI (dark blue) was analyzed by whole cell staining of PRY514. Bar indicates 2 μm.

The online version of this article includes the following source data and figure supplement(s) for figure 3:

**Source data 1.** Source data for *Figure 3*.

**Source data 2.** Source data for *Figure 3*.

**Figure supplement 1.** *MPS3-MER2S-Flag* shows wild-type spore viability.

---

TSS; *Figure 3E*). We purified a GST fusion protein of an Mps3 fragment (residues 160–260), including TSS. In vitro Cdc28–Clb5 alone, but not Cdc7–Dbf4 alone, promoted phosphorylation of the Mps3 fragment (approximately 10-fold compared to the background; *Figure 3F*). Co-incubation with CDK and DDK resulted in robust incorporation of $^{32}$P in GST-Mps3, indicating that CDK and DDK cooperate in Mps3 phosphorylation in vitro (~30 -fold compared to the background). Triple alanine substitutions (T188A, S189A, S190A; referred to as the *mps3-AAA* mutant; *Figure 3E*) made the GST-Mps3 fragment a poor substrate by these kinases. These results indicate that TSS is required for efficient CDK/DDK phosphorylation in vitro.

Consistent with the in vitro results, the Mps3-AAA mutant protein showed a reduced band shift (*Figure 3B and G*) and decreased reactivity to anti-phosphoserine antibody (*Figure 3H*). Furthermore, we raised an antibody against a Mps3 peptide with 189-phosphoserine and 190-phosphoserine (186-GAT**pSpS**PGKSF-195) and found that this phospho-specific antibody could recognise immunoprecipitated Mps3-Flag specifically between 4 and 8 h in meiosis, but not in mitosis (*Figure 3B*, left panel). Strong signals were observed during late prophase I (4  and 6 h). This antibody did not recognise precipitated Mps3-AAA-Flag (*Figure 3B*, right panel). These results indicate that 189 S and 190 S are phosphorylated in vivo, specifically during meiosis.

The Mps3 sequence around the TSS site is similar to that of CDK/DDK-catalysed phosphorylation sites in Mer2 (*Henderson et al., 2006*; *Sasanuma et al., 2008*; *Wan et al., 2008*), including the TSSP sequence (*Figure 3E*). Mer2 is a meiosis-specific protein essential for DSB formation (*Cool and Malone, 1992*; *Engebrecht et al., 1990*). CDK and DDK sequentially phosphorylate the TSS sequence of Mer2, which is essential for meiotic DSB formation (*Sasanuma et al., 2008*; *Wan et al., 2008*). We swapped a 10-amino-acid region (GATSSPGKSF) containing the Mps3 phosphorylation sites with the corresponding region of Mer2 (TETSSPFRST; *Figure 3E*). This *MPS3* allele (*MPS3-MER2S*) conferred wild-type spore viability (95 %; n = 50 tetrads; *Figure 3—figure supplement 1*). Importantly, the Mps3-MER2S protein cross-reacted with an antibody against the Mer2 phospho-S30 antibody (*Henderson et al., 2006*; *Figure 3I*). *MPS3-MER2S-Flag* exhibited similar telomere (Rap1) clustering to the wild type (*Figure 3J and K*). The Mps3-MER2S-Flag protein showed meiosis-specific

localisation on NE which often co-localized with Ndj1 (*Figure 3L*). This suggests that residues 26–35 of Mer2 are functionally equivalent to residues 185–194 of Mps3. Rather, the sequence of Mps3, the composition of amino acids, seems to be critical for NE localisation. Similar to Mps3, the region of Mer2 is rich in serine and threonine (*Figure 3E*).

## The *Mps3-AAA* mutant is defective in Mps3 localization on NE

To investigate the role of the phosphorylated residues of Mps3, we constructed and characterised the phenotypes of the *mps3-AAA-FLAG* mutant (hereafter, *mps3-AAA*; *Figures 3E* and *4*). The *mps3-AAA* mutant showed few defects in mitosis, such as SPB duplication (*Figure 4—figure supplement 1*) seen in other *mps3* mutants (*Jaspersen et al., 2002*; *Nishikawa et al., 2003*), suggesting that the TSS residues of Mps3 are not critical for the mitotic function of Mps3. The *mps3-AAA* mutant showed defects in meiosis with a reduced spore viability of 87.5 % (*Figure 4A*), which was slightly higher than that of the *ndj1* mutant (75.8%) (*Chua and Roeder, 1997*; *Conrad et al., 1997*). The spore viability of the *ndj1 mps3-AAA* double mutant was 72.3%, which is similar to that of the *ndj1* mutant. The *mps3-AAA* mutant delayed entry into meiosis I by ~1 h relative to the wild type (*Figure 4A*). The *ndj1 mps3-AAA* double mutant showed ~2.5 h delayed entry into meiosis I, which is similar to the *ndj1* single mutant, indicating the epistatic relationship of *mps3-AAA* to *ndj1*. The *mps3-AAA* mutant showed normal progression of the meiotic S-phase (*Figure 4—figure supplement 1*). Immunostaining of Zip1, a component of the central region of SC (*Sym et al., 1993*), showed a delay in both the loading of Zip1 and the formation of full-length SCs in the *mps3-AAA* mutant (*Figure 4—figure supplement 1*), suggesting that Mps3 NE localisation is necessary for timely synapsis.

We also analysed chromosome dynamics during meiosis in *mps3-AAA* mutant cells. First, we checked the Rap1-GFP movement in the *mps3-AAA* mutant. In contrast to wild-type cells, the *mps3-AAA* mutant was defective in the localisation and motion of Rap1-GFP (*Figure 4B and C*). In the mutant, the number of Rap1 foci did not increase (*Figure 4B*), and little movement of Rap1-GFP foci was observed during meiosis (*Figure 4C*). Rap1 motion in meiotic prophase I of the *mps3-AAA* mutant is restricted, and the velocity of each Rap1 focus is decreased relative to the wild type (*Figure 4C and D* and *Figure 4—video 1*). Therefore, we concluded that T188, S189, and S190 of Mps3 are important for meiotic telomere dynamics. Indeed, the *mps3-AAA* mutant was defective in chromosome motion, as shown by the analysis of a GFP fusion of Zip1 (*Koszul et al., 2008*; *White et al., 2004*). During prophase I, Zip1-GFP showed dynamic motion (*Figure 4E*; *Figure 4—videos 2 and 3*). The *mps3-AAA* mutant reduced the velocity of Zip1 motion by approximately threefold relative to that of the wild type (48 ± 15 vs 130 ± 31 nm/s, *Figure 4F*).

Next, we examined the localisation of the Mps3-AAA mutant protein in meiosis as a GFP fusion protein (*Figure 4G*). *mps3-AAA-GFP* cells showed 85 % spore viability, which was comparable to the mutant strain without the GFP tag. Similar to the wild-type Mps3-GFP, the Mps3-AAA GFP fusion protein normally resided on the SPB as a single focus during mitosis (*Figure 4G*). During meiosis, the Mps3-AAA mutant protein showed reduced NE localisation. Upon entry into meiosis, similar to the wild-type Mps3, the Mps3-AAA mutant protein formed a few (usually two or three small) foci on NE, but showed little increase in the number of Mps3 foci. At late time points, unlike wild-type Mps3, the mutant protein did not form an Mps3 patch or cover on NE (*Figure 4G, H,I*). At 6 h, two or three foci of Mps3 showed limited movement (*Figure 4J, K and L*; *Figure 4—video 4*). These results suggest that T188, S189, and S190 of Mps3 are necessary for meiosis-specific Mps3 localisation and motion on NE, particularly for efficient localisation on NE accompanied by the formation of large complexes. Interestingly, the Mps3-AAA-Flag mutant protein still bound to Ndj1 and Csm4, similar to the wild-type Mps3 protein (*Figure 4—figure supplement 1*), which is consistent with the Mps3-AAA protein retaining the N-terminal nucleoplasmic region and SUN domain necessary for Ndj1 and Csm4 binding, respectively.

## The *Mps3-AAA* mutant is deficient in NE growth

In addition to localization/movement defects, the *mps3-AAA* mutant showed defects in nuclear morphology. First, the meiotic nucleus of the *mps3-AAA* mutant was smaller than that of the wild type (*Figure 4B*). The nucleus of the wild type grew when the cell underwent meiosis and exhibited a fourfold increase (in area; twofolds in length) at 4 h (*Figure 4M*). Treatment with LatB did not affect the expansion, indicating that meiosis-specific NE remodelling does not require actin polymerisation.

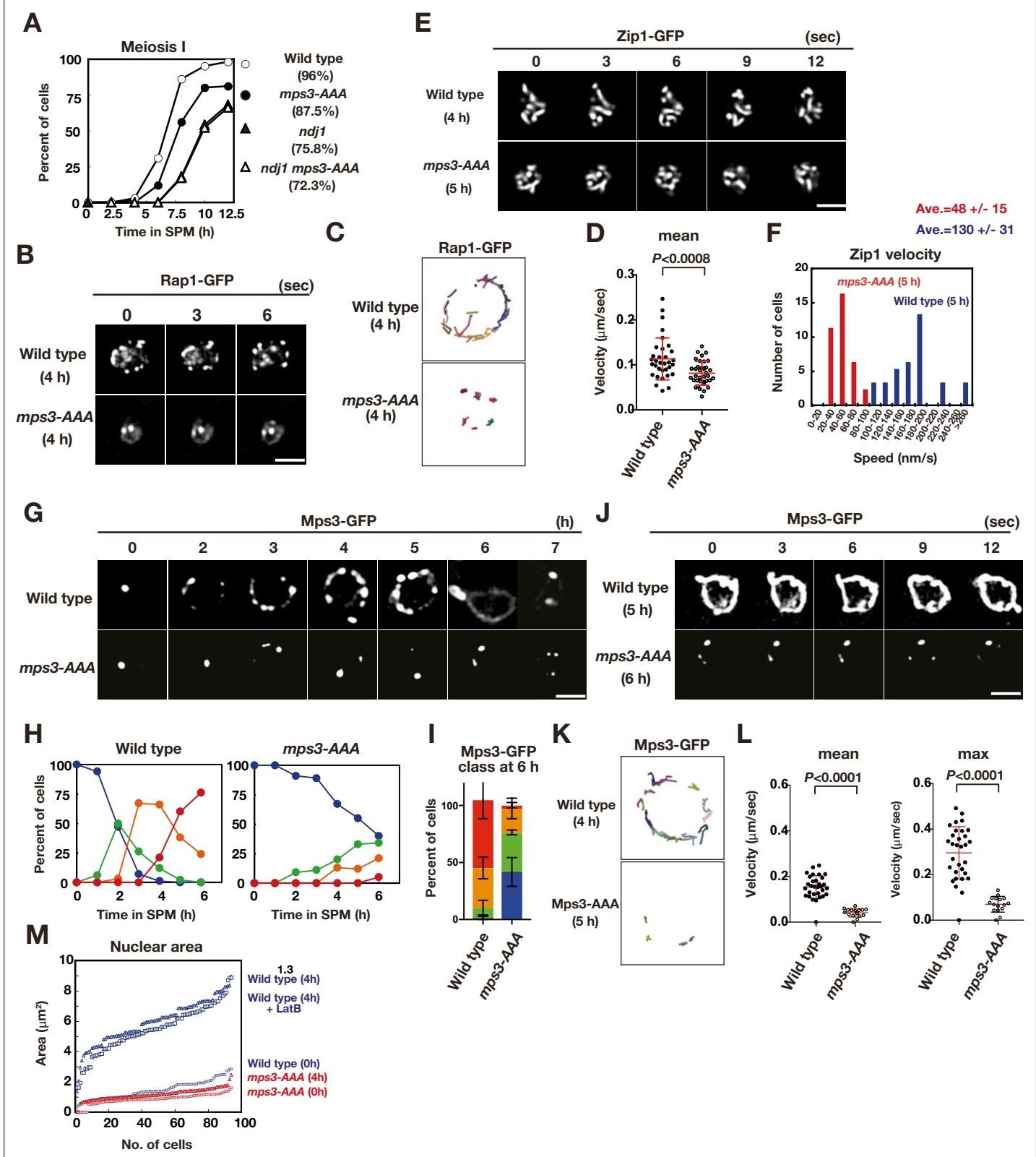

**Figure 4.** The *mps3-AAA* mutant impairs telomere and NE expansion. (**A**) Meiosis division was analyzed by DAPI staining for various strains. More than 100 cells were counted for divisions. The graphs are a representative of two independent time course. Spore viability of each strain is shown in parenthesis. At least 100 asci were dissected for each strain. (**B**) Time-lapse analysis of Rap1-GFP in the wild-type (HKY167) and *mps3-AAA* (PRY138) strains at different time points in meiosis. A single focal plane of a cell was analyzed every 3 s. A tethering defect was observed in the *mps3-AAA*

*Figure 4 continued on next page*

*Figure 4 continued*

strains. Bar indicates 2 µm. See *Figure 4—video 1* for *mps3-AAA*. (**C**) Tracking of Rap1-GFP in wild type and *mps3-AAA* for 20 s. See Materials and methods in *Figure 2B*. (**D**) Velocity of Rap1-GFP foci or patches were quantified. See Materials and Methods in *Figure 2C*. Red lines show mean with s.d. p-Values were calculated using Man-Whitney's *U*-test. (**E**) Time-lapse analysis of Zip1-GFP in wild type (SEY672) and *mps3-AAA* (PRY332). *mps3-AAA*. See *Figure 4—videos 2 and 3*. (**F**) Tracing of Zip1-GFP shows a step size of each Zip1-GFP line. Step size per given time is converted into the relative velocity of chromosomes. n = 35. Blue bars; wild type; red bars, *mps3-AAA*. (**G**) Localization of wild-type Mps3-GFP (PRY64) and Mps3-AAA-GFP (PRY186) proteins in a cell at different time points in meiosis. (**H**) Kinetics of Mps3 distribution. Based on Mps3-GFP morphology, cells with Mps3-GFP were classified into four classes and quantified: 2–5 foci (green), more than five foci/patches (orange), and coverage on NE (red). See *Figure 1B* for quantification. (**I**) Percentages of cells with different classes of Mps3-GFP were quantified at 5 h in different conditions (triplicates, Error bars show standard deviation; s.d.); single Mps3 focus (blue), 2–5 foci (green), more than five foci/patches (orange), and coverage of the Mps3 signal on NE (red). (**J**) Time-lapse analysis of wild type and Mps3-AAA-GFP during meiosis. See *Figure 4—video 4* for Mps3-AAA-GFP. (**K**) Tracking of Mps3-GFP in wild type and *mps3-AAA* for 20 s. See *Figure 1F* for quantification. (**L**) Velocity of Mps3-GFP foci or patches were quantified. Methods are shown in *Figure 1H*. More than 20 cells were analyzed for the quantification. Red lines show mean with s.d. p-Values were calculated using Man-Whitney's *U*-test. (**M**) The maximum sectional area of each nucleus was measured using the Velocity program. Each area of the 96 nuclei is ranked in the figure. Graphs for wild type and *mps3-AAA* at 0 and 4 h as well as wild type treated with LatB are shown.

The online version of this article includes the following video, source data, and figure supplement(s) for figure 4:

**Source data 1.** Source data for Figure 4.

**Figure supplement 1.** CDK and DDK are necessary for Mps3 localization on NE.

**Figure 4—video 1.** Rap1-GFP in *mps3-AAA* cells at 5 h in SPM.

https://elifesciences.org/articles/63119/figures#fig4video1

**Figure 4—video 2.** Zip1-GFP in a wild-type cell at 5 h in SPM.

https://elifesciences.org/articles/63119/figures#fig4video2

**Figure 4—video 3.** Zip1-GFP in *mps3-AAA* cells at 6 h in SPM.

https://elifesciences.org/articles/63119/figures#fig4video3

**Figure 4—video 4.** Mps3-AAA-GFP at 5 h in SPM.

https://elifesciences.org/articles/63119/figures#fig4video4

The *mps3-AAA* mutant showed defective NE growth in prophase I; the nuclear area in the *mps3-AAA* mutant was similar between 0 and 4 h (*Figure 4M*). These results indicate that the Mps3 luminal region plays a critical role in the growth of the meiotic nucleus, possibly by controlling NE remodelling.

Second, a fraction of the *mps3-AAA* mutant cells showed a partial defect in telomere tethering to NE during meiosis. In one or two Rap1-GFP foci, the *mps3-AAA* mutant (40 %; n = 100) was localised to the nucleoplasm rather than to the nuclear periphery, whereas all Rap1-GFP foci were localised near the NE in the wild type (*Figure 4B*). The weak telomere tethering defect in the *mps3-AAA* mutant is reminiscent of the defect observed with an *NDJ1* mutant, which also showed a partial tethering defect (*Conrad et al., 1997*; *Trelles-Sticken et al., 2000*).

## Aspartate substitution of TSS in Mps3 overcomes the CDK and DDK defects

We constructed a version of Mps3 with three negative charges on the TSS sequence (T188D, S189D, and S190D), referred to as *MPS3-DDD* (*Figure 3E*). The *MPS3-DDD* allele did not show any mitotic defects, such as colocalization with tubulin and tubulin elongation (*Figure 5—figure supplement 1*). The *MPS3-DDD* cells showed wild-type spore viability (95%) and entered MI similar to wild-type cells (*Figure 5—figure supplement 1*). The *MPS3-DDD* cells also displayed similar Rap1-GFP dynamics to wild-type cells with normal clustering during prophase I (*Figure 5A and B*). The Mps3-DDD-GFP protein localised to SPB during vegetative growth (*Figure 5—figure supplement 1*) and showed wild-type-like localisation on NE during meiosis (*Figure 5C and D*, *Figure 5—figure supplement 1*). Mps3-DDD-GFP displayed movement similar to that of the wild-type protein (*Figure 5C, E and F*; *Figure 5—video 1*). The aspartate substitutions of Mps3 suppressed the resolution defect in Rap1 clustering at late time points of meiosis by CDK and DDK inactivation (*Figure 5A and B*), although some clustering persisted under these conditions (*Figure 5B*). Mps3-DDD-GFP showed NE localisation (*Figure 5D*) and movement during meiosis even under the conditions of either CDK or DDK inactivation (*Figure 5E and F*; *Figure 5—videos 2–5*), which is similar to that in wild-type Mps3-GFP or Mps3-DDD-GFP cells. Clustering of Mps3 induced by CDK/DDK inactivation was clearly ameliorated

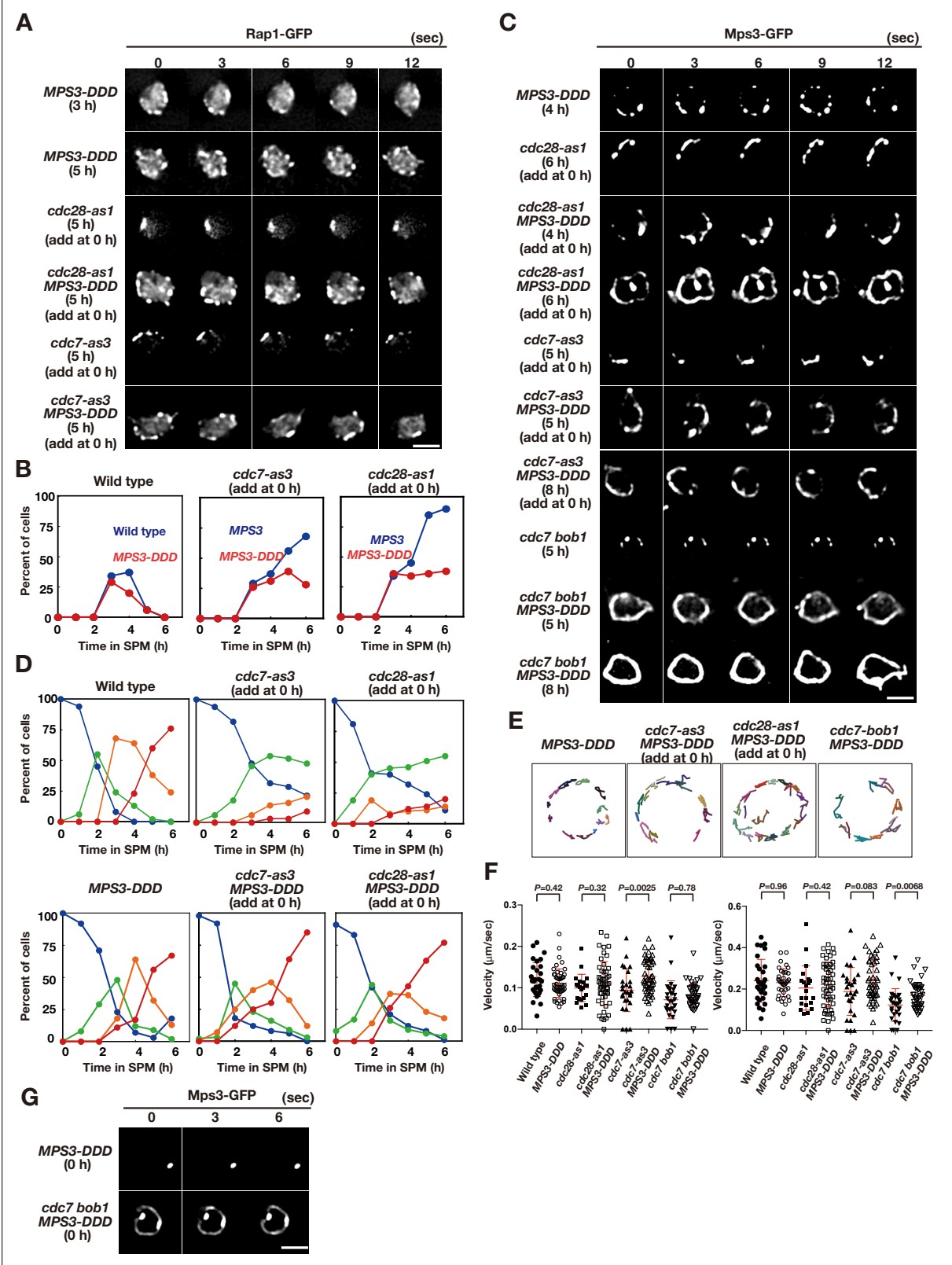

**Figure 5.** Acid amino acids in lumen of Mps3 can suppress CDK and DDK defects. (**A**) Time-lapse analysis of Rap1-GFP in a phosphomimetic allele of *MPS3*, *MPS3-DDD* (PRY211), with a different combination of *cdc28-as1* (PRY211), *cdc7-as3* (PRY301), and *cdc7 bob1-1* (PRY272) mutations in meiosis. A single focal plane of a cell was analyzed at every 3 s. Bar indicates 2 μm. See *Figure 5—videos 2–5*. (**B**) Kinetics of clustering of telomere as Rap1-clusters was studied in *cdc7-as3* (left, PRY309) and *cdc28-as1* (right, PRY303) in the presence of its specific inhibitors. Wild type (blue, HKY167) and

*Figure 5 continued on next page*

*Figure 5 continued*

*MPS3-DDD* (PRY236) with the allele was used. (**C**) Time-lapse analysis of Mps3-DDD-GFP with different mutant alleles and in treatment with a specific inhibitor at different time points in meiosis. See *Figure 5—videos 1 and 6–8*. (**D**) Kinetics of Mps3. Based on Mps3-GFP morphology, cells with Mps3-GFP were classified into four classes and quantified: single Mps3 focus (blue), 2–3 foci (green), 4–5 foci (orange), and more than five foci (red). See *Figure 1B* for quantification. Graphs for wild type, *cdc28-as1*, and *cdc7-as3* cells are the same as in *Figure 1B*. (**E**) Tracking of Mps3-GFP in *MPS3-DDD*, *cdc28-as1 MPS3-DDD* (with 1NM-PP1), *cdc7-as3 MPS3-DDD* (with PP1), and *cdc7 bob1-1MPS3-DDD* for 20 s. (**F**) Velocity of Mps3-GFP foci or patches were quantified. Methods are shown in *Figure 1H*. More than 20 cells were analyzed for the quantification. Red lines show mean with s.d. p-Values were calculated using Man-Whitney's *U*-test. (**G**) Time-lapse analysis of Mps3-DDD-GFP with different mutant alleles during vegetative growth.

The online version of this article includes the following video, source data, and figure supplement(s) for figure 5:

**Source data 1.** Source data for *Figure 5*.

**Figure supplement 1.** Meiotic phenotypes of *MPS3-DDD* mutants.

**Figure 5—video 1.** Mps3-DDD-GFP at 5 h in SPM.
https://elifesciences.org/articles/63119/figures#fig5video1

**Figure 5—video 2.** Rap1-GFP in *MPS3-DDD* cells at 5 h in SPM.
https://elifesciences.org/articles/63119/figures#fig5video2

**Figure 5—video 3.** Rap1-GFP in a *cdc28-as1 MPS3-DDD* cell treated with 1NM-PP1 (0 h) at 5 h in SPM.
https://elifesciences.org/articles/63119/figures#fig5video3

**Figure 5—video 4.** Rap1-GFP in a *cdc7-as3 MPS3-DDD* cell treated with PP1 (0 h) at 5 h in SPM.
https://elifesciences.org/articles/63119/figures#fig5video4

**Figure 5—video 5.** Rap1-GFP in a *cdc7 bob1-1 MPS3-DDD* cell at 5 h in SPM.
https://elifesciences.org/articles/63119/figures#fig5video5

**Figure 5—video 6.** Mps3-DDD -GFP in a *cdc28-as1 MPS3-DDD* cell treated with 1NM-PP1 (0 h) at 5 h in SPM.
https://elifesciences.org/articles/63119/figures#fig5video6

**Figure 5—video 7.** Mps3-DDD -GFP in a *cdc7-as3 MPS3-DDD* cell treated with PP1 (0 h) at 5 h in SPM.
https://elifesciences.org/articles/63119/figures#fig5video7

**Figure 5—video 8.** Mps3-DDD -GFP in a *cdc7 bob1-1 MPS3-DDD* cell at 5 h in SPM.
https://elifesciences.org/articles/63119/figures#fig5video8

by the introduction of Mps3-DDD-GFP (*Figure 5C and E*; *Figure 5—videos 6–8*). DDD substitution also suppressed the motion defects of Mps3 by DDK inactivation (*Figure 5F*). Importantly, NE localisation and dynamics of Mps3-DDD protein still depend on meiosis, suggesting the presence of additional regulation of meiosis-specific localisation of Mps3. In vegetative cells (0 h, *Figure 5—figure supplement 1*), Mps3-DDD showed SPB localisation and normal Rap1 localisation (*Figure 5G*, *Figure 5—figure supplement 1*). Surprisingly, Mps3-DDD, but not wild-type Mps3, showed dispersed staining on NE with SPB in mitotic cells only in the *cdc7 bob-1* mutant with, but not in *cdc7as3* with the inhibitor (*Figure 5H*, *Figure 5—figure supplement 1*). This suggests that, in contrast to meiosis, DDK negatively regulates the NE localisation of Mps3 in vegetative cells.

## The *mps3-S189A* and *mps3-S190A* mutants phenocopy the *Mps3-AAA* mutant

Although the *MPS3-DDD* mutation suppressed Mps3 localisation defects induced by CDK or DDK inactivation, Mps3 localisation defects in the *mps3-AAA* mutant (reduced localisation) are different from those under CDK/DDK inactivation (persistent clustering). To clarify the relationship between the phosphorylation of the luminal region of Mps3 with CDK and DDK, we created additional mutants in the Mps3 TSS sequence, called *mps3-S189A and -S190A*, which has a single alanine substitution of S189 and S190, respectively (*Figure 6A*), which are phosphorylated in vivo (*Figure 3*). If CDK could phosphorylate S190 as the CDK site, *mps3-S190A* would phenocopy the CDK defect in Mps3 localisation. And also, if DDK could phosphorylate S189 as the DDK site, *mps3-S189A* would phenocopy the DDK defect in Mps3 localisation. The localisation of GFP fusion of Mps3-S189A and -S190A proteins was analysed as described above (*Figure 6B and C*). Both Mps3-S189A and -S190A proteins showed reduced NE localisation compared to wild-type Mps3. In the localisation defects, both mutants were weaker than the *mps3-AAA* mutant (*Figure 6D*). The *mps3-S189A* mutant was more severe than the *mps3-S190A* mutant. Both *mps3* mutants also showed reduced Mps3 motion (*Figure 6E and*

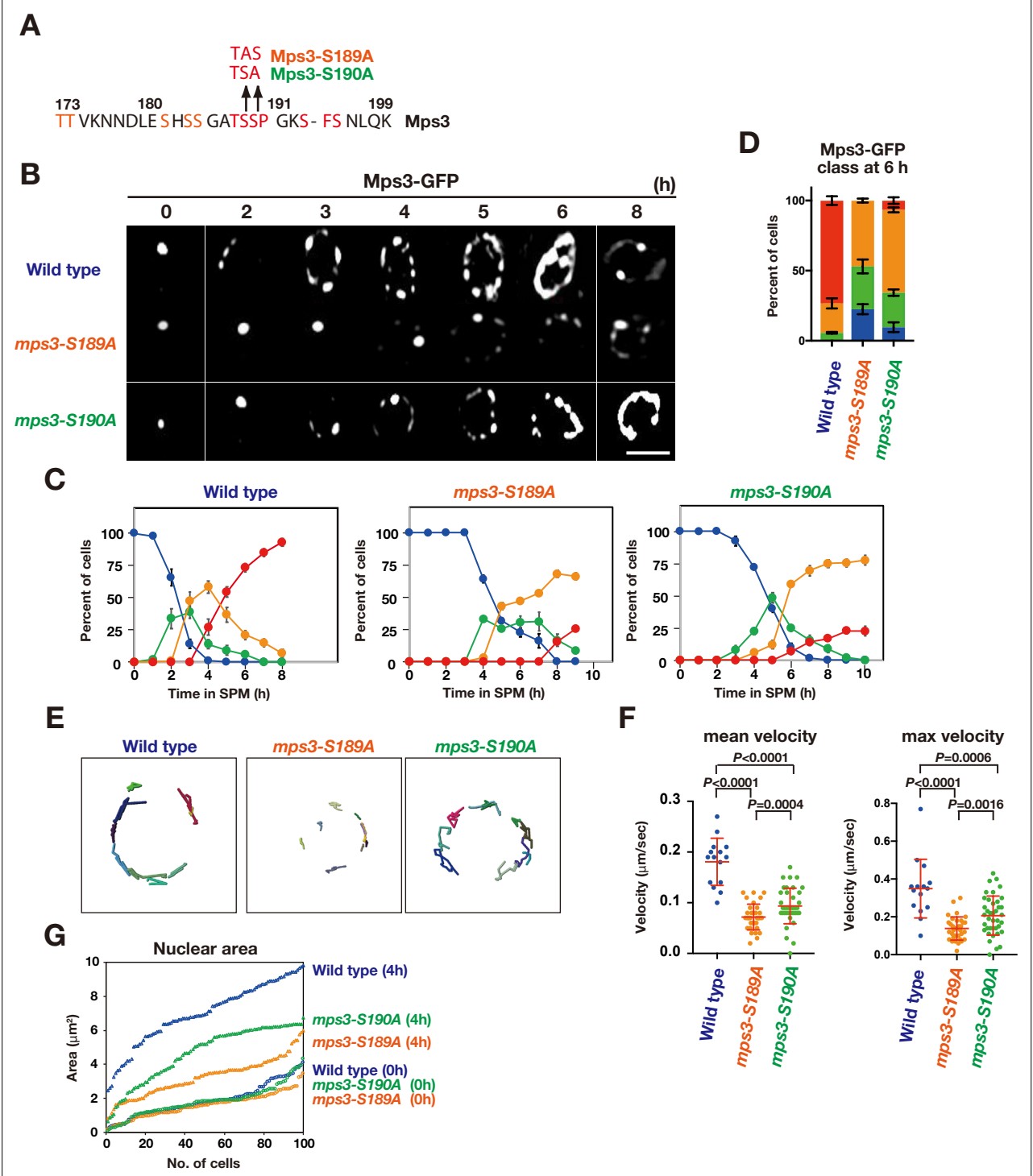

**Figure 6.** The *mps3-S189A* and *-S190A* mutants show a defect in NE localization. (**A**) Amino acid sequences of Mps3 with amino acid substitution in the *mps3-S189A* and *mps3-S190A* mutants. (**B**) Localization of wild-type Mps3-GFP (PRY64), Mps3-S189A-GFP (KSY407/409) Mps3-S190A-GFP (KSY220/221) and proteins in a cell at different time points in meiosis. (**C**) Kinetics of Mps3 distribution. Based on Mps3-GFP morphology, cells with Mps3-GFP were classified into four classes and quantified: 2–5 foci (green), more than five foci/patches (orange), and coverage on NE (red). See *Figure 1B* for quantification. (**D**) Percentage of cells with different classes of Mps3-GFP were quantified at 5 h in different conditions (triplicates, Error bars show standard deviation; s.d.); single Mps3 focus (blue), 2–5 foci (green), more than five foci/patches (orange), and coverage of the Mps3 signal on NE (red). (**E**) Tracking of Mps3-GFP in wild type (left), *mps3-S189A* (middle) and *mps3-S190A* (right). (**F**) Velocity of Mps3-GFP foci or patches were quantified in different *mps3* mutants. Methods are shown in *Figure 1H*. More than 20 cells were analyzed for the quantification. Red lines show mean with s.d.

*Figure 6 continued on next page*

*Figure 6 continued*

p-Values were calculated using Man-Whitney's *U*-test. (**G**) The maximum sectional area of each nucleus was measured using the Velocity program. Each area of the 96 nuclei is ranked in the figure.

The online version of this article includes the following source data for figure 6:

**Source data 1.** Source data for *Figure 6*.

*F*) and were also partially deficient in meiosis-induced nuclear expansion (*Figure 6G*). The *mps3-S190A* mutant exhibited weaker defects than the *mps3-S189A* mutant. This suggests that, although weaker, the *mps3-S189A* and *-S190A* mutants shared similar defects to the *mps3-AAA* mutant and were different from *cdk* or *ddk* mutants. Taken together, these results suggest that neither S189 nor S190 are directly phosphorylated by CDK or DDK in vivo (see Discussion).

## Luminal region of Mps3 binds to lipid bilayers in vitro

To determine the role of putative phosphorylation of the Mps3 luminal region, we analysed the biophysical properties of a 55 amino acid Mps3 peptide (153–208 aa) containing both transmembrane (TM) and luminal regions (*Figure 3A*). The luminal region is referred to as the juxtamembrane (JM) region (*Figure 7A*). We prepared two peptides, Mps3-wild type (WT) and Mps3-DDD, which contain three aspartates (*Figure 7A*). Synthesised Mps3 peptides were mixed with liposomes comprising POPC/POPS (10/3)(*Sato et al., 2006*) First, we checked the conformation of each peptide in the bilayers by polarised Fourier-transform infrared spectroscopy (FT-IR) and found that both Mps3-WT and Mps3-DDD peptides contained similar contents of α-helices and β-strands (*Figure 7—figure supplement 1*). Dichroic ratio calculated for Mps3-WT and Mps3-DDD were 2.90 and 2.85, suggesting that both transmembrane regions are inserted into lipid bilayers with a similar tilt angle of ~30° relative to the membrane normal (*Tamagaki et al., 2014*). We cannot assign those peaks from the β-strand to which residues in the JM regions are involved in the β-structure with this FT-IR experiment alone.

Next, we synthesised Mps3 TM-JM peptide with Alexa fluorophore-dye (Alexa-568) at the C-terminus of the JM region and analysed the spectrum of the dye in the presence of different concentrations of negatively charged phosphatidyl-inositol 4,5-bisphosphate (PIP$_2$), which modulates the status of charge on the membrane surface (*Sato et al., 2009*). Here, we utilised a PIP$_2$-involving model system to determine whether JM is affected by the membrane, although we are aware of the fact that it is not known whether Msp3 interacts with PIP$_2$. This simple model system might provide a possible mechanism of how negative charge on the JM region caused by its phosphorylation affects its behaviour against the membrane.

For the wild-type peptide, the addition of PIP$_2$ decreased the peak intensity of the fluorophore at 600 nm up to ~40 % (*Figure 7B*). The triplicates showed similar results. Decreased fluorescence could be due to quenching of the fluorescence from the dye association (self-quenching) by inter-peptide association (*Figure 7C*), as shown in a previous study on an ErbB2 peptide containing TM and JM regions (*Matsushita et al., 2013*). These results suggest that the JM region of the Mps3-WT peptide is sensitive to changes in the membrane component and that the JM regions interact with the acidic lipid bilayer. On the other hand, the Mps3-DDD peptide decreased the intensity to ~20 % (half of the intensity from the WT peptide). The decrease in the change in the fluorescence intensity was due to electrostatic repulsion between the aspartates and acidic lipids (*Figure 7C*). This repulsion must place the fluorophores apart, preventing self-quenching. The repulsion can reduce the JM region of the Mps3-DDD peptide binding to the bilayers compared to that of Mps3-WT. These results suggest the different conformations of Mps3-WT and -DDD peptides with respect to the interaction of the JM region of Mps3 with the membrane.

To gain more insight into the behaviour of the JM region with the membrane, we utilised a molecular dynamic (MD) simulation technique. Since there is no reported structure on the TM-JM region of Mps3, we artificially constructed the initial structure of the TM-JM sequence as follows: using the software *Chimaera* (*Pettersen et al., 2004*), R161-M181 is set to be α-helix as the TM region; the rest was set to a random structure. This constructed structure was embedded in a membrane (POPC/POPS/PIP$_2$ = 8:2:1 with 80 nm in XY dimension) using *the Membrane Builder* module in *CHARMM-GUI* (*Jo et al., 2008*). The peptide-membrane system was fully hydrated in a 150 mM NaCl solution.

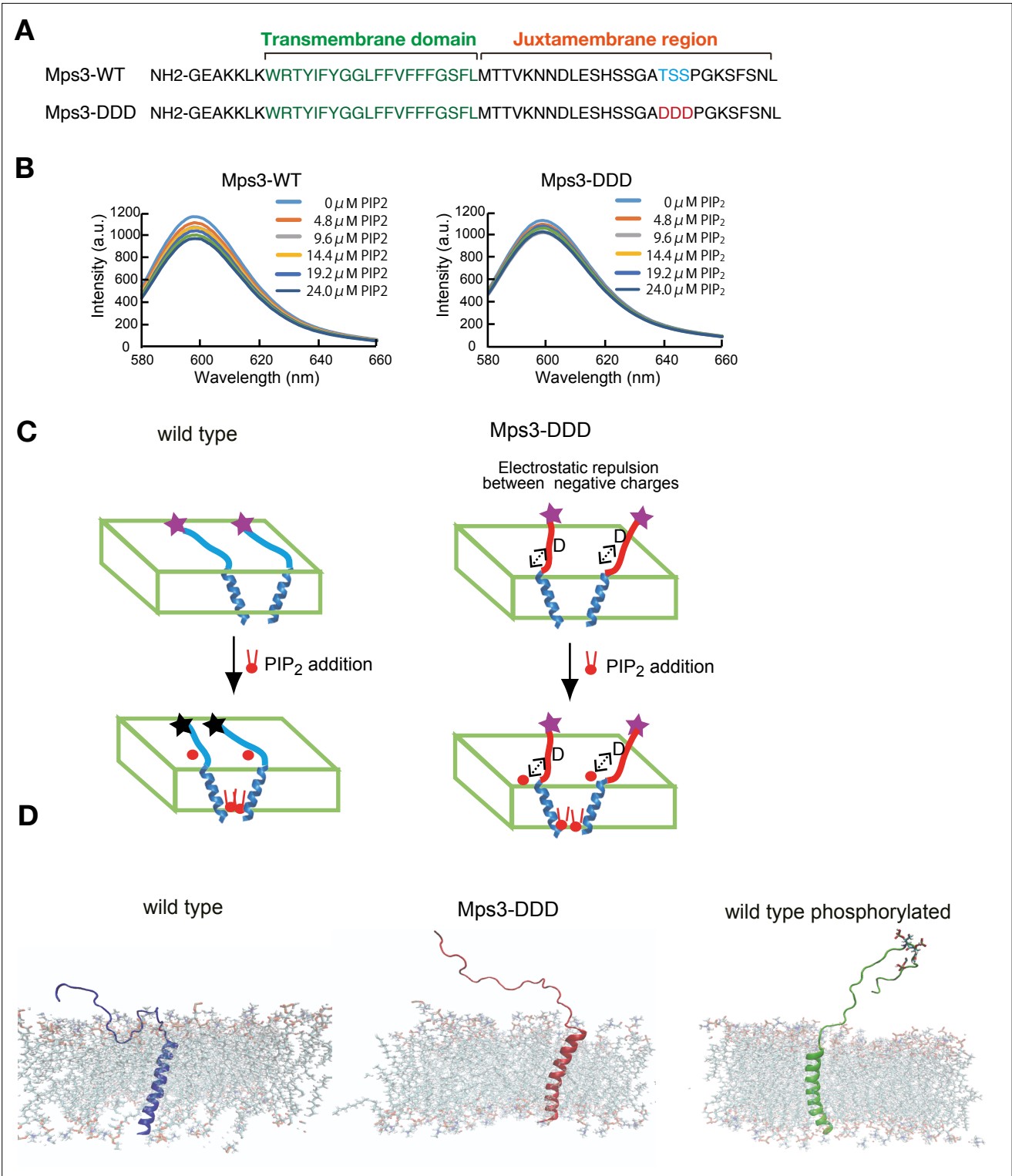

**Figure 7.** Biophysical analysis of JM region of Mps3. (**A**) Amino acid sequences of Mps3 peptides used. Amino acid sequence in transmembrane region is shown in green. Phosphorylated residues TSS and its derivative DDD are shown in blue and red, respectively. (**B**) Fluorescence spectra from a synthetic liposome containing Alexa568-labelled Mps3-WT (left) peptide and Mps3-DDD peptide (right) were measured in the presence of various concentrations of PIP$_2$. The graphs are a representative of three independent time course. (**C**) Schematic illustration of the binding of Mps3-WT peptide (left) and Mps3-DDD peptide (right) to lipid bilayer with PIP$_2$ (red). DDD residues may induce electric repulsion between the peptide and the membrane. Transmembrane region of the Mps3 peptide is shown by blue a-helix. Peptides with WT JM region and with DDD substitution are shown blue and red lines, respectively. Purple and black stars show normal and quenched fluorophores, respectively. (**D**) Snapshots from all-atom MD simulation trajectories

*Figure 7 continued*

for Mps3-WT peptide (left), Mps3-DDD peptide (center) and Mps3-WT peptide with phosphorylation at TSS sequence (right). See *Figure 7—videos 1–3*.

The online version of this article includes the following video and figure supplement(s) for figure 7:

**Figure supplement 1.** Structure and dynamics of JM region of Mps3.

**Figure 7—video 1.** MD simulation of wild-type Mps3 peptide.

https://elifesciences.org/articles/63119/figures#fig7video1

**Figure 7—video 2.** MD simulation of wild-type Mps3-DDD peptide.

https://elifesciences.org/articles/63119/figures#fig7video2

**Figure 7—video 3.** MD simulation of wild-type Mps3-phosphorylated peptide.

https://elifesciences.org/articles/63119/figures#fig7video3

First, we performed coarse-grained molecular dynamics simulations of the constructed TM-JM peptide in the membrane. We ran simulations on two sequences, Mps3-WT and Mps3-DDD. The duration of each run was >3 μs, and three sets were performed for each sequence. The results from all the calculations showed that the JM region from Mps3-DDD dissociated from the membrane (*Figure 7—figure supplement 1*). Conversely, that of Mps3-WT remained attached to the membrane (*Figure 7—figure supplement 1*). Second, we performed all-atom simulations on three TM-JM sequence peptides, Mps3-WT, Mps3-DDD, and Mps3-WT phosphorylated at T188-S189-S190. The duration of each calculation was >50 ns (*Figure 7—videos 1–3*). In *Figure 7D*, snapshots from trajectories for the three sequences are shown. Results from the simulation, together with mass density profiles (*Figure 7—figure supplement 1*), indicate that JM regions from Mps3-DDD and phosphorylated Mps3 dissociated from the membrane. On the other hand, the JM region from Mps3-WT remained attached to the membrane.

## Discussion

Here, we describe the multi-layered control of the dynamics of an SUN-domain protein, Mps3, during yeast meiosis. In both mitotic and meiotic cells, Mps3 is a major component of a half-bridge as well as the SPB interacting network (SPIN) (*Chen et al., 2019*; *Lee et al., 2020a*; *Li et al., 2017*). Meiosis induces the localisation and motion of Mps3 on NE, which is accompanied by the formation of large protein ensembles of the LINC complex in NE, detected as foci/patches. The meiosis-specific LINC complex tethers telomeres and interacts with cytoplasmic actin cables (*Lee et al., 2020b*). In the luminal region between INM and ONM, Mps3 binds to a KASH protein, Mps2, during both mitosis and meiosis, and to a meiosis-specific Mps2-binding protein, Csm4 (*Chen et al., 2019*; *Lee et al., 2020a*). Mps2 and Csm4 cooperate to bind to the motor Myo2 in the cytoplasm. Myo2 on actin cables, whose formation is induced in meiotic cells, generates forces for the motion of the LINC complex on NE, which in turn moves chromosomes inside of meiotic cells (rapid prophase movement; RPM) as well as the clustering of telomeres in the zygotene stage (*Conrad et al., 2008*; *Koszul et al., 2008*; *Trelles-Sticken et al., 2000*; *Trelles-Sticken et al., 1999*). While Mps3 localisation on NE is independent of actin-generated force (and other meiosis-specific factors, such as Ndj1 and Csm4), Mps3 movement on NE is dependent on the force. Meiosis-specific regulation of Mps3 localisation on NE is a key event in the formation of a LINC complex with Mps3 for chromosome pairing/synapsis and chromosome motion.

Our studies revealed that, in addition to its role in chromosome dynamics, Mps3 localisation in NE is important for NE remodelling during meiosis. The characterisation of the *mps3-AAA* and *mps3-S189A* and *-S190A* mutants revealed that Mps3 localisation to NE (NE remodelling) plays a unique role in NE expansion in meiotic cells. NE-bound Mps3 may regulate NE growth by directly promoting lipid synthesis on NE or by importing lipids from the ER (*Sosa Ponce et al., 2020*).

Localisation of Mps3 on NE in meiotic cells is divided into several distinct steps, which are coupled with distinct regulatory mechanisms (*Figures 1A and 8A*); 1-At early times in prophase I, in addition to the SPB, a few Mps3 foci appear on NE. 2-With increased numbers of foci, patches of Mps3, large cohesive ensembles, probably with multiple foci, are formed on NE, which shows transient clustering in one area of NE. 3-Multiple foci and patches cover the entire NE. 4-Mps3 shows nearly full coverage

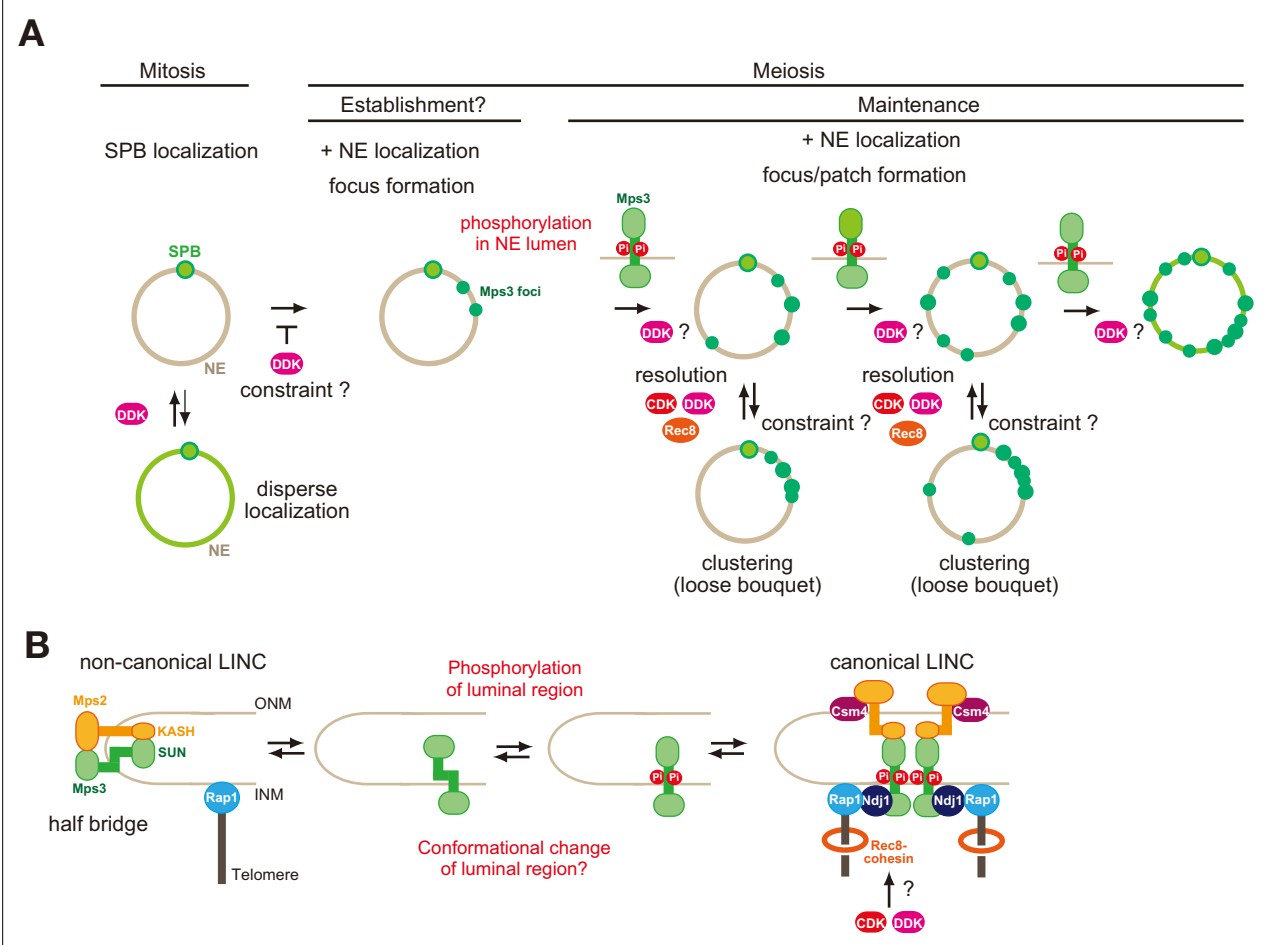

**Figure 8.** A model of regulation of Mps3 on NE during meiosis. (**A**) Multiple phosphorylation regulates Mps3 localisation and motion in NE during meiosis. In mitotic cells, Mps3 (green) is mainly located in the SPB. In some strain, disperse weak Mps3 signal is detected on NE, which might be regulated by DDK. Upon entry into meiosis, Mps3 forms a few foci on NE in early prophase I, and phosphorylation of the luminal region of Mps3 and, probably DDK, promotes localisation of more Mps3 proteins as focus/patch on NE. Mps3 foci/patches transiently form a cluster of some Mps3 foci by an unknown mechanism, and CDK and DDK and Rec8-cohesin promote dissociation of Mps3 and Rap1(telomere) clusters. It is likely that positive feedback on Mps3 phosphorylation would increase the localisation of Mps3 on NEs, which results in full NE coverage of Mps3. (**B**) A hypothetical model of the formation of Mps3-containing LINC complex. During mitosis, Mps3 forms a non-canonical LINC complex with *cis*-membrane interactions with Mps2 as a component of the half-bridge and SPIN (left). Upon induction of meiosis, unknown factors promote the formation of Mps3 foci on NE. Then, the JM region of Mps3 is subject to non-canonical phosphorylation, which might in turn weaken the binding of JM to INM, probably by electric repulsion between the JM regions with negative charges and acid lipids. This process promotes the formation of the canonical LINC complex with the *trans*-membrane configuration, in which the Mps3 SUN domain binds to the KASH domain of Mps2 and Csm4. Csm4 may promote structural changes in the luminal region of Mps2. The N-terminal region of Mps3, which is located in the nucleoplasm, binds to a telomere-binding protein, Ndj1. During middle and late prophase I, Mps3 forms a large protein ensemble on the NE, which is seen as a patch.

of NE, sometimes with a hole at the nucleus-vacuole junction (NVJ). 5-During late prophase I or pro-metaphase I, most Mps3 on NE disappear from NE, leaving two SPB-associating foci. At each stage, Mps3 foci/patches show RPM which is the maximum at mid/late prophase I (*Conrad et al., 2008*; *Koszul et al., 2008*; *Trelles-Sticken et al., 2000*; *Trelles-Sticken et al., 1999*).

In this study, we found that two cell cycle kinases, CDK and DDK, control proper Mps3 localisation on NE during meiosis, particularly by promoting the resolution of clustered Mps3 foci (and, thus, telomere bouquets), suggesting that the phosphorylation of a target protein(s) by these kinases plays a role in the resolution of Mps3 clusters. Moreover, we showed that the JM region of Mps3 near the INM, which is located in the lumen of NEs, is critical for meiosis-specific NE localisation of the protein. S179 and S180 of the JM region in Mps3 are subject to meiosis-specific phosphorylation of Mps3 assembly on NEs. The introduction of negative charges in the JM region changes the binding ability of

the region to the membrane in reconstituted liposomes. These results suggest that Mps3 localisation in NE is subject to multi-layered regulation governed by distinct phosphorylation.

## Can CDK and DDK phosphorylate the lumen of Mps3?

During meiosis of *C. elegans*, the N-terminal nucleoplasmic region of Sun-1 protein is phosphorylated by Chk-1 and Cdk-1 (*Penkner et al., 2009*; *Zuela and Gruenbaum, 2016*). In M phase of human cells, SUN1 protein is phosphorylated at its N-terminal region by CDK and PLK (*Patel et al., 2014*). Together with results described here, these suggest the importance of the phosphorylation in the regulation of dynamics and/or regulation of SUN-domain proteins.

Our in vitro study (*Figure 3F*) showed that CDK and DDK phosphorylate 188-TSS-190 sequence of Mps3 in a collaborative manner, suggesting that CDK and DDK collaborate to catalyze the phosphorylation of the JM region of Mps3 for its NE localization in vivo. However, this is less likely for several reasons. First, it is unlikely that CDK and DDK are responsible for direct phosphorylation reaction in NE lumen. These kinases are not localized to luminal regions of both endoplasmic reticulum (ER) and NE since the catalytic subunits (Cdc28 and Cdc7) and its regulatory subunits (Clb5,6 and Dbf4) lack a leader sequence necessary for the luminal localization. Second, defects in Mps3 localization induced by CDK/DDK inactivation are different from those in the *mps3-AAA* mutant. Third, if CDK is a priming kinase on S190 in SSP sequence of Mps3, a mutation in the CDK site (S190) could phenocopy the defect conferred by CDK defect. But this is not the case (*Figure 6*). Fourth, if DDK is the secondary kinase on S-pS-P primed by CDK, S189 substitution would confer similar (or less) defect to S190 substitution. However, the *mps3-S189A* is more defective than the *mps3-S190A*. Rather, S190 and S189 phosphorylation seem to play an overlapping role in Mps3 localization.

## What is the target for CDK and DDK in Mps3 localization?

If CDK and DDK are unlikely to directly phosphorylate the luminal region of Mps3, what are the target(s) of these kinases? Our results indicate that CDK and DDK activities promote the resolution of Mps3 foci/patches and/or telomeres rather than NE localisation per se. Mps3 localisation on NE seems normal, even in the absence of CDK and DDK activities. Phenotypic similarity between *cdk* and *ddk* mutants in Mps3 localization suggests a common target of these kinases for the resolution of the clusters. Interestingly, telomere-resolution defects under CDK and DDK inactivation are similar to those in the mutant of the *REC8* gene (*Challa et al., 2016*; *Conrad et al., 2007*), which encodes a meiosis-specific kleisin subunit of the cohesin complex (*Klein et al., 1999*) and cohesin regulator, *RAD61/WPL1* (*Challa et al., 2016*). It is well known that Rec8 is phosphorylated by DDK (and Polo and CK1 kinases), but not by CDK (*Katis et al., 2010*) and Rad61/Wpl1 is phosphorylated by DDK and Polo-like kinase (PLK) (*Challa et al., 2019*). Thus, there might be another target in cohesin components by the CDK-DDK kinase axis. In mitotic yeast cells, Eco1, which catalyses the acetylation of the Smc3 cohesin subunit, is phosphorylated by CDK and DDK for degradation (*Seoane and Morgan, 2017*). Interestingly, Eco1 acetylates Mps3 in vegetative cells (*Ghosh et al., 2012*). Therefore, Eco1 may be a possible target of CDK-DDK for telomere resolution. Alternatively, there might be as yet unknown substrates by these kinases for the Mps3 dynamics.

Telomere clustering during prophase I persisted with CDK or DDK inactivation, suggesting the presence of a constraint on the resolution of the clustering. The phosphorylation, and thus negative charges in the JM region, might overcome this constraint for resolution. This is supported by the fact that *MPS3-DDD* partially rescued the resolution defect of telomere clustering with CDK or DDK inactivation (*Figure 4*). Moreover, we found that Mps3-DDD, but not wild-type Mps3, showed NE localisation in mitotic cells only in the *cdc7 bob-1* mutant (*Figure 5C*), suggesting the presence of DDK-dependent constraints on the NE localisation of Mps3 during mitosis (*Figure 8A*).

## Phosphorylation in the NE luminal region of Mps3

Protein phosphorylation of the luminal region between INM and ONM is very unusual and has not been reported in the past. However, recent studies have shown that the phosphorylation of secreted proteins and oligosaccharides of membrane proteins is catalysed by an atypical kinase, such as Family with sequence similarity 20, member C (Fam20C), in the lumen of the Golgi apparatus and ER in vertebrate cells (*Tagliabracci et al., 2012*; *Tagliabracci et al., 2013*; *Tagliabracci et al., 2015*). Given that the luminal region of NE is consistent with that of ER, where INM/ONM proteins are synthesised, it

is possible that a Fam20C family-like kinase might be present in the lumen between INM and ONM, which may catalyse the phosphorylation of the NE luminal region of Mps3. However, there are no reports on the presence of Fam20C orthologues or relative enzymes in lower eukaryotes, such as yeasts. Alternatively, phosphorylation of the luminal region may be mediated by a novel mechanism. Given that membranes are rich in phospho-lipids, the phosphate group in the lipids might act as a donor for transfer to proteins. Further studies are needed to identify the protein responsible for the phosphorylation of the JM region of Mps3.

## The role of negative charges in JM region of Mps3

Biophysical analysis of synthetic peptides containing TM and JM regions of Mps3 in synthetic liposomes revealed that the JM region of Mps3 has the capability to bind to the lipid bilayers. Importantly, the introduction of three negative charges into the JM region weakens the binding to the membrane, resulting in a change in the geometry of the JM region of Mps3, as an extension, the luminal region of Mps3, including the SUN domain. As a typical LINC complex, the SUN domain of Mps3 in INM binds to the KASH domain of Msp2 in ONM. The connection spans ~20 nm between the INM and ONM (*Chen et al., 2019*; *Lee et al., 2020a*). Recently, Jasperson and her colleagues showed "atypical" configuration of Mps3-Mps2 in SPB, in which not only SUN-KASH interaction, but also N-terminal regions of both Mps3 and Mps2 bind to each other in SPIN which surrounds SPB (*Chen et al., 2019*; *Lee et al., 2020a*). Typical LINC could accommodate *trans*-membrane configuration in the luminal region, while atypical LINC would have *a cis*-membrane configuration (*Figure 8B*). This suggests that the luminal regions of Mps3 and Mps2 are quite flexible in accommodating two different SUN-KASH configurations. Our results suggest that the switch from *cis*- to *trans*-membrane configurations of the LINC complex might be regulated by the binding of the JM region of Mps3 to the INM. While the interaction between the JM region of Mps3 and INM promotes the *cis*-membrane form of the noncanonical LINC complex, negative charges induced by phosphorylation may reduce the interaction between JM and INM, which may accommodate a conformation suitable for the *trans*-membrane form. Alternatively, the negatively charged JM region of Mps3 would be a binding site for the other protein which may promote the configuration switch. This idea should be tested in future studies.

## Meiosis establishes Mps3 localization on NE

As described above, Mps3 localisation in NE is regulated at distinct stages (*Figure 8*). The *mps3-AAA* mutant retains the ability to form early meiosis-specific Mps3 foci on NE. The Mps3-DDD protein also requires meiosis for NE localisation during meiosis. This suggests that Mps3 localisation on NE is initiated and/or established during very early meiotic prophase I. Given that Mps3 localisation as an ensemble on NE is specific to meiosis, there might be an early meiosis-specific factor or modification that regulates the establishment of Mps3 localisation on NE upon entry into meiosis. We examined two meiosis-specific factors, Ndj1 and Csm4, both of which are involved in chromosome motion and Mps3 dynamics, but it is unlikely that these factors function in the establishment of Mps3 localisation. Further studies are necessary to identify the factors necessary for the initiation of NE remodelling.

# Materials and methods

Key resources table

| Reagent type (species) or resource | Designation | Source or reference | Identifiers | Additional information |
|---|---|---|---|---|
| Strain, strain background (*Saccharomyces cerevisiae*) | SK1 | PMC246633 | | |
| Antibody | Anti-HA (mouse monoclonal) | Babco | 16B12 | WB (1/1000) |
| Antibody | Anti-Flag (M2) (mouse monoclonal) | Sigma | M2 | WB (1/1000), IF (1/2000) |
| Antibody | Anti-tubulin (rat monoclonal) | Bio-Rad/Serotec, Ltd | MCA77G | WB (1/1000), IF (1/2000) |
| Antibody | Anti-Zip1 (rabbit polyclonal) | PMID:18297071 | | IF (1/200) |

*Continued on next page*

*Continued*

| Reagent type (species) or resource | Designation | Source or reference | Identifiers | Additional information |
|---|---|---|---|---|
| Antibody | Anti-Csm4 (rabbit polyclonal) | PMC2533704 | | WB (1/1000) |
| Antibody | Anti-Mer2(S30P) (rabbit polyclonal) | PMC2216698 | | WB (1/1000) |
| Antibody | Anti-Mps3(S189P, S190P) (rabbit polyclonal) | This study | | WB (1/500) |
| Chemical compound, drug | 1NM-PP1 | Cayman Chemical | Cat #: 13,330 CAS-221244-14-0 | |
| Chemical compound, drug | PP1 | Sigma | Cat #:0040 CAS-172889-26-8 | |
| Chemical compound, drug | Latrunculin B | Calbiochem | Cat-428020 | |
| Software, algorithm | Imaris | Oxford Instruments | | Tracking/speed measurement |

## Strains and plasmids

All strains described here are derivatives of SK1 diploids and are described in *Supplementary file 1*. *MPS3-FLAG* and *MPS3-GFP* diploid cells showed wild-type spore viability, but both showed ~2 h delay in the entry of meiosis I (*Rao et al., 2011*). *RAP1-GFP* and *NDJ1-HA* diploids showed wild-type spore viability and normal progression of meiosis (*Kosaka et al., 2008*; *Rao et al., 2011*; *Trelles-Sticken et al., 1999*). GST fusion genes of a PCR-amplified Mps3 fragment (160–260 amino acid residues of the Mps3 protein) were constructed on pGEX-2T (Cytiva, 28954653).

## Anti-serum and antibodies

Anti-HA antibody (16B12; Babco), anti-Flag (M2, Sigma), and anti-tubulin (MCA77G, Bio-Rad/Serotec, Ltd) were used for western blotting and/or imuno-staining. Anti-Zip1 (rabbit) and anti-Csm4 (rabbit) were described previously (*Kosaka et al., 2008*; *Shinohara et al., 2008*). The second antibodies for staining were Alexa-488 (Goat) and −594 (Goat) IgG used at a 1/2000 dilution (Themo Fishers).

Ten amino acids of Mps3 peptide with 189-phosphoserine and 190-phosphoserine (186-GAT**pSpS**PGKSF-195) was synthesized and used for the immunization of rabbit. From obtained serum, antibody specific phosphor-peptide was affinity-purified using the phosphor-peptide (MBL Co. Ltd. Japan).

## Cytology

Time-lapse images of Mps3-GFP, Rap1-GFP and Zip1-GFP were captured using a computer-assisted fluorescence microscope system (DeltaVision; Applied Precision). The objective lens was an oil immersion lens (100×; NA, 1.35). Image deconvolution was performed using an image workstation (Soft-Works; Applied Precision). Time-lapse image acquisition was performed every 0.5 to 1 s at a single focal plane. Meiotic cell cultures (~150 µl) were placed on bottom-glass dishes (Matsunami) precoated with Concavalin A (10 mg/ml, Sigma).

Tracking of Zip1-GFP/Mps3-GFP/Rap1-GFP was analyzed using the ImageJ program (NIH) and plug-in MTrack2 (Nico Shuuman). Tracking and velocity analysis of Mps3-GFP/Rap1-GFP were analyzed using Imaris software (Oxford Instrument). The sectioned area of each nucleus with Rap1-GFP, which showed nuclear and dotty staining, was measured using the Velocity program (Applied Precision) with manual assignment of the rim of the sectioned area. Ninety-six nuclei were counted.

## IP and western blotting

Yeast cell lysates were prepared by the glass bead disruption method. The lysates were incubated with magnetic beads (Dynal M260; GE Healthcare) coated with anti-Flag antibody (M2, Sigma) for 12 h and washed extensively (*Sasanuma et al., 2013*). Bound proteins were eluted by adding the SDS sample buffer and were analyzed on an SDS-PAGE gel, transferred to a nylon membrane (Millipore Co. Ltd), and probed with specific antibodies.

## In vitro kinase assay

For in vitro phosphorylation, 1 µg of purified GST-Mps3 or GST-Mps3-AAA was incubated with $\gamma$-$^{32}$P-ATP (100 µM; 2.2 MBq) in the presence of CDK (~0.2 µg) and/or DDK (~0.2 µg) in a 20 µl of

buffer [20 mM Tris-HCl (pH 7.5), 10 mM MgCl$_2$, 10 mM β-glycerophosphate]. After 1 h incubation at 30 °C, the reaction was stopped by adding 5× SDS PAGE sample buffer, and the products were analyzed on 10 % SDS-PAGE gel. Dried gels were analyzed using a phosphorimager BAS 2000 (Fuji Co. Ltd).

## Synthesis of Mps3 peptides

Peptides with the TM and JM region of Mps3 (residue 153–208) were synthesized by solid-phase method with the following sequence: GEAKKLKWRTYIFYGGLFFVFYFFGSFLMTTVKNNDLESHSSGA**TSS**PGKSFSNL-Cys and GEAKKLKWRTYIFYGGLFFVFYFFGSFLMTTVKNNDLESHSSGA**DDD**PGKSFSNL-Cys (underlines indicate TM region). The synthetic peptides were purified by reverse-phase HPLC on a C4 column with gradient of 2-propanol over water containing 0.1 % trifluoroacetic acid. For fluorescent-labeling, Alexa Fluor 568 C5-maleimide (Molecular probe) was introduced to the sulfide group on cysteine by mixing the peptides and the fluorescent dye in dimethylformamide under basic conditions.

## Preparation of lipid bilayer with peptides

The Mps3 peptides were co-solubilized with 1-palmitoyl-2-oleoyl-*sn*-phosphatidycholine (POPC), 1-palmitoyl-2-oleoyl-*sn*-phosphatidyserine (POPS) and octyl-β-glucoside in trifluoroethanol (TFE). The peptide to lipid molar ratio was 1:100. The lipid concentration was 200 µM. The ratio of POPC and POPS was 10:3. The mixture was incubated for 90 min at 37 °C. Then TFE was removed under stream of Ar. MES buffer (10 mM MES, 50 mM NaCl, and 5 mM DTT, pH 6.2) was added to the solid from the previous step and mixed at 37 °C for 6 h. The octyl-β -glucoside was removed by dialysis against the MES buffer.

## Measurement of FT-IR

Polarized attenuated total reflection FT-IR spectra were obtained on a JASCO FT/IR-4700 spectrometer. Membranes containing Mps3 TM–JM peptides were layered on a germanium internal reflection element using a slow flow of nitrogen gas directed at an oblique angle to the IR plate to form an oriented multilamellar lipid–peptide film.1000 scans were acquired and averaged for each sample at a resolution of 4 cm$^{-1}$. The absorption of polarized light by the amide I bond yields the dichroic ratio defined as a ratio of absorption intensity for parallel, relative to perpendicular, polarized light. From the dichroic ratio, we estimated the tilt angle of the TM helix relative to the membrane normal based on the method described by Smith and coworkers (*Liu et al., 2004*; *Smith et al., 2002*) using a value of 41.8° for angle α between the helix director and the transition-dipole moment of the amide I vibrational mode. For the FT-IR experiment, the amount of lipid used per experiment was ~4 mg. The film on the ATR plate in our experiment was assumed to be greater than ~10 µm. For calculating the dichroic ratio, the thick film limit is applicable (*Bechinger et al., 1999*). Equations that we used for the calculation of the dichroic ratios were based on this assumption.

## Fluorescence spectroscopy

For experiments with PIP$_2$, TM-JM peptides were inserted into POPC/POPS membrane as explained above. The peptide/lipid ratio was set to 1:1000 for observing the fluorescence from Alexa568. The lipid concentration was 200–250 µM in MOPS buffer (1 mM MOPS, 0.1 M KCl, pH 7.0). Vesicles were formed by extrusion of multilamellar vesicles through 200 nm polycarbonate filters. PIP$_2$ were added into the membranes by addition of PIP$_2$ micelles to the vesicle solution. Fluorescence was measured within 1 hour to minimize PIP$_2$ hydrolysis. The PIP$_2$ concentration was from 5 µM to 25 µM. Fluorescence measurement was carried out on HITACHI F-2500 fluorescence spectrophotometer at the excitation wavelength of 568 nm.

## Coarse-grained MD simulation

All simulations were performed using GROMACS 2016 package (*Abraham et al., 2015*) using the MARTINI force field with polarizable water (*Marrink et al., 2007*). Periodic boundary conditions were applied in all directions, and the time step was 20 fs. The temperature was controlled with the Berendsen temperature coupling scheme (*Berendsen et al., 1984*) with a time constant of 1 ps, and the pressure was controlled using the Parrinello–Rahman semi-isotropic barostat (*Parrinello and*

*Rahman, 1981*) with a time constant of 12 ps and a compressibility of $3 \times 10^{-4}$ bar $^{-1}$. All the simulations were carried out in the isothermal–isobaric (NPT) ensemble at a temperature of 303 K and pressure of 1.0 bar. The cut-off of the Van der waals interactions were set to 1.1 nm.

### All atom MD simulation

The NAMD software package (*Phillips et al., 2005*) was used to perform all-atom MD simulations. The CHARMM36m force field (*Huang et al., 2017*) together with TIP3P water model were used for all simulations. All simulations in NPT dynamics (constant particle number, pressure and temperature) were carried out at 303 K with Langevin dynamics, and at one atm maintained by Nosè-Hoover Langevin piston method. Full electrostatic interactions were treated by the particle mesh Ewald (PME) approach with a grid spacing of less than 1 Å. The cut-off radii of long-range electrostatic and van der Waals interactions were set to be 12 Å, with a smoothing function applied from 10 Å.

## Acknowledgements

We are grateful to Drs. Nancy Hollingsworth, Franz Klein, Kunihiro Ohta, Kazu Nishikawa, Hisao Masai, and Hiroyuki Araki for providing the materials used in this study. We thank Ms. Saeko Hashimoto for excellent technical assistance. HBDPR was supported by the BMC program and a scholarship from the Graduate School of Science, Osaka University, JASSO as well as MEXT. KC were supported by the Institute for Protein Research. This work was supported by a Grant-in-Aid from the JSPS KAKENHI Grant Number; 22125001, 22125002, 15H05973,16H04742, and 19H00981 to AS.

## Additional information

### Funding

| Funder | Grant reference number | Author |
|---|---|---|
| Japan Society for the Promotion of Science | 22125001 | Akira Shinohara |
| Japan Society for the Promotion of Science | 22125002 | Akira Shinohara |
| Japan Society for the Promotion of Science | 15H05973 | Akira Shinohara |
| Japan Society for the Promotion of Science | 16H04742 | Akira Shinohara |
| Japan Society for the Promotion of Science | 19H00981 | Akira Shinohara |

The funders had no role in study design, data collection and interpretation, or the decision to submit the work for publication.

### Author contributions

Hanumanthu BD Prasada Rao, Conceptualization, Data curation, Formal analysis, Investigation, Methodology, Visualization, Writing – review and editing; Takeshi Sato, Conceptualization, Data curation, Formal analysis, Investigation, Resources, Visualization, Writing – review and editing; Kiran Challa, Data curation, Formal analysis, Investigation, Visualization, Writing – review and editing; Yurika Fujita, Methodology, Software, Writing – review and editing; Miki Shinohara, Methodology, Resources, Writing – review and editing; Akira Shinohara, Conceptualization, Data curation, Formal analysis, Funding acquisition, Investigation, Supervision, Validation, Visualization, Writing - original draft, Writing – review and editing

### Author ORCIDs

Akira Shinohara (iD) http://orcid.org/0000-0003-4207-8247

### Decision letter and Author response

Decision letter https://doi.org/10.7554/eLife.63119.sa1

Author response https://doi.org/10.7554/eLife.63119.sa2

## Additional files

### Supplementary files
• Transparent reporting form
• Supplementary file 1. Strain list.

### Data availability
The numerical data in all Figures (graphs) are provided in Source data. Original blots and gels are provided in Source data.

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
