## [Decision Letter]

Thank you for submitting your article "Phosphorylation of luminal region of the SUN-domain protein Mps3 promotes nuclear envelope localization during meiosis" for consideration by *eLife*. Your article has been reviewed by 3 peer reviewers, and the evaluation has been overseen by a Reviewing Editor and Kevin Struhl as the Senior Editor. The following individual involved in review of your submission has agreed to reveal their identity: Owen Richard Davies (Reviewer #1).

The reviewers have discussed the reviews with one another and the Reviewing Editor has drafted this decision to help you prepare a revised submission.

As the editors have judged that your manuscript is of interest, but as described below that additional experiments are required before it is published, we would like to draw your attention to changes in our revision policy that we have made in response to COVID-19 (https://elifesciences.org/articles/57162). First, because many researchers have temporarily lost access to the labs, we will give authors as much time as they need to submit revised manuscripts. We are also offering, if you choose, to post the manuscript to bioRxiv (if it is not already there) along with this decision letter and a formal designation that the manuscript is "in revision at eLife". Please let us know if you would like to pursue this option. (If your work is more suitable for medRxiv, you will need to post the preprint yourself, as the mechanisms for us to do so are still in development.)

Summary

The manuscript by Rao et al., examines the phosphorylation of *S. cerevisiae* LINC Mps3 and provides evidence that phosphorylation is important for Mps3 meiotic nuclear envelope (NE) localization. Based on biophysical analyses and molecular modeling, the authors present a model that Mps3 phosphorylation alters interaction with membranes for production of a canonical LINC complex, critical for chromosome pairing and synapsis. The authors specifically show: (1) that proper Mps3 NE localization, chromosome movement and telomere clustering is dependent on CDK and DDK kinases; (2) sequences within the Mps3 lumenal domain are phosphorylated and CDK/DDK can phosphorylate these sequences in vitro; (3) mutations that block this phosphorylation result in the predicted meiotic defects; (4) mutations to phosphomimetic amino acids result in a Mps3 version that is localized to the NE to promote chromosome movement and telomere clustering independently of CDK and DDK; (5) in vitro liposome analyses and molecular modeling suggest altered interaction with membranes when the domain is phosphorylated (carries negative charges). Based on this in vitro data, the authors present a model whereby phosphorylation drives formation of a canonical LINC complex in trans to promote NE localization, which in turn is important for chromosome movement and telomere clustering. The finding of a functionally important meiosis-specific phosphorylation on Mps3 will be of interest in the field and conclusions are largely supported by experimental data. However, further experiments are required to support the idea that this region is actively controlling the localization of Msp3 in the manner the authors suggest in the model. Extensive improvements are also required to the text to ensure clarity and brevity.

Essential revisions

1. This manuscript studies the nuclear localisation and movement of Mps3 and Rap1 by showing images at 3-second time intervals at various stages during prophase I. Whilst the authors state clear differences in movement, these are not always obvious from the figures, especially when mutants are studied at 5hr, when WT movements are not as apparent. The confidence in these data would be strengthened if the movements were quantified (across multiple nuclei) and statistical analysis performed of the WT and test populations. This could be achieved using the tracking data (e.g. Figure 1F) or by measuring correlations between static images at 3-second intervals. Furthermore, more details on the methodology are required and more cells need to be analysed for the estimated average speed and distance including statistical analysis (Lines 722-723 "Zip1-GFP / Mps3-GFP /Rap1-GFP)…".

2. Although it is clear that the removal of cdc28 and cdc7 through the use of mutants sensitive to the analogs impacts the localization of Mps3, it is important to check that the amount of Mps3 protein remains stable. The authors state that the effect is only on the localization, but the stability of Mps3 may be affected and this may cause the phenotype. The analysis of the phosphomutant Mps3-AAA in Supplementary Figure 3, for example, it is not sufficient to rule this out. Indeed, a similar analysis of Mps3-GFP must be performed under cdc28-as1 and cdc7-as3 inhibition to visualize how the loss of this kinase activity affects the stability of Mps3.

3. Phosphorylation of Mps3: The authors show data that Mps3 is phosphorylated in a CDK and DDK dependent manner. This includes phenotypic analyses of inactivated kinases for Mps3 NE localization, chromosome movement and telomere clustering, direct phosphorylation of the Mps3 luminal domain by these kinases in vitro, and normal NE localization in the phosphomimetic Mps3 mutant in the cdk/ddk background. The authors do not look at the phosphorylation of Mps3 in vivo in the inactivated kinase backgrounds. However, in the discussion, the authors discuss the likelihood that CDK and DDK are NOT responsible for phosphorylating Mps3 due to the luminal localization. This point needs to be clarified with a better explanation for the role of CDK and DDK. At a minimum, the authors should indicate on page 10 that "... although CDK/DDK can phosphorylate Mps3 in vitro they are unlikely to phosphorylate Mps3 in vivo (see discussion)."

4. How much of the Mps3 protein is phosphorylated? This is relevant to the generation of the Mps3-DDD version and the finding that there are no mitotic defects. The authors should be able to resolve this point by quantitative western blotting in the different conditions. It is also unclear based on the provided images whether Mps3-DDD still localizes to the SPB – this should be addressed.

5. The biophysical data of Figure 6B are interpreted as showing that DDD peptides exhibit increased repulsion between aspartate side-chains of different peptides and between aspartate side-chains and acidic lipids (lines 475-477). Shouldn't an increase in repulsion between side-chains result in increased fluorescence in absence of PIP2, owing to a reduction in self-quenching? This isn't apparent in the data, in which DDD appears to have slightly lower fluorescence than WT at 0 uM. The second conclusion of increased repulsion between side-chains and acidic lipids appears reasonable and is supported by the MD data. This should be addressed by providing further data that support the first conclusion, or removing it and focussing solely on the second conclusion.

6. The authors present a model of the interaction between SUN-KASH domains of the LINC complex based on in vitro liposome experiments in combination with molecular modelling. This is the novel part of the manuscript and if correct is an important finding. While the manuscript is focused on the role of phosphorylation in changing its dynamics, rather than the architecture of the LINC complex directly, figure 7 needs to be revised to show exactly what is known and what is speculation. Ideally, it would be desirable to substantiate the model with in vivo evidence, for example by super-resolution imaging. However, it is understood that the experiments are technically challenging as the NE is 20-nm wide so picking out this architecture by super-resolution microscopy would be a significant achievement. In addition, to make the mechanistic claim that phosphorylation is necessary for the interaction to occur (as is suggested in the figure) analysis of Mps2-Mps3 interaction in vivo in the presence of phosphorylation mutants would be required. The authors should clarify the caveats of their model in figure 7, either with appropriate changes to the text, or by the provision of additional data to support it.

7. The text needs extensive revision. It is frequently difficult to follow and contains numerous errors, mis-labeled figures, omitted error bars etc. – it should be carefully edited for grammar and to improve clarity and precision. The discussion is unusually long and includes much published information that is not necessary for contextualising the findings. It also needs substantial editing to maximise clarity and brevity. Some are listed below, but this is not an exhaustive list.

---

## [Author Response]

Essential revisions1. This manuscript studies the nuclear localisation and movement of Mps3 and Rap1 by showing images at 3-second time intervals at various stages during prophase I. Whilst the authors state clear differences in movement, these are not always obvious from the figures, especially when mutants are studied at 5hr, when WT movements are not as apparent. The confidence in these data would be strengthened if the movements were quantified (across multiple nuclei) and statistical analysis performed of the WT and test populations. This could be achieved using the tracking data (e.g. Figure 1F) or by measuring correlations between static images at 3-second intervals. Furthermore, more details on the methodology are required and more cells need to be analysed for the estimated average speed and distance including statistical analysis (Lines 722-723 "Zip1-GFP / Mps3-GFP /Rap1-GFP)…".

The motion of Mps3 and Rap1 foci/patches on NE is very heterogeneous and very much complicated to track down. As previously shown (Conrad et al., 2008; Lee et al., 2012), the motion analysis even for a single locus marked with GFP such as a telomeric locus showed very complicated patterns of the motion with at least two (or three) distinct classes of maximam velocity. To avoid complicated analysis of the motion, we measured a motion velocity of multiple Mps3-GFP particles in a single focal plane, using the simple tracking analysis in the Imaris software (Oxford Instrument) and presented mean and maximum speeds of each particle of Mps3-GFP (Methods and statistics are shown in Materials and methods). The results are shown in revised Figures 1H, 4L, 6E and S1D for Mps3-GFP, and also in Figures 2C and 4D for Rap1-GFP. More importantly, to show the dynamics nature of the motion, we also provided videos of a representative dynamics of Mps3-GFP and Rap1-GFP in different strains under various conditions (Videos 1-25). It is very clear that the *mps3-3A* as well as *cdc7* deletion (with *bob1-1*) significantly reduces the speeds of Mps3 foci. In addition, interestingly, CDK inactivation showed slight reduction of the velocity of the Mps3 motions; thus, it seems that CDK plays a major role in the resolution of Mps3 foci/patches, but little in the motion (as discussed in our previous version). These are clearly mentioned in the text. Moreover, we replaced tracking results in Figures 1F, 2B, 5H and S1B, in which each focus tracking is shown in different colors (for 20 s tracking instead of previous 120 s tracking).

We do not include data for tracking distances of Mps3(Rap1)-GFP focus since the distance along which the GFP focus moves depends on duration time of the focus; each focus showed distinct duration time even within 20 second tracking time.

2. Although it is clear that the removal of cdc28 and cdc7 through the use of mutants sensitive to the analogs impacts the localization of Mps3, it is important to check that the amount of Mps3 protein remains stable. The authors state that the effect is only on the localization, but the stability of Mps3 may be affected and this may cause the phenotype. The analysis of the phosphomutant Mps3-AAA in Supplementary Figure 3, for example, it is not sufficient to rule this out. Indeed, a similar analysis of Mps3-GFP must be performed under cdc28-as1 and cdc7-as3 inhibition to visualize how the loss of this kinase activity affects the stability of Mps3.

We performed western blotting analysis of Mps3-Flag in CDK (*cdc28-as1*) and DDK (*cdc7-as3*) inactivation conditions, which is provided in Supplemental Figure S1E and S1G. Compromised CDK and DDK activities did not affect steady state levels of Mps3-Flag protein.

3. Phosphorylation of Mps3: The authors show data that Mps3 is phosphorylated in a CDK and DDK dependent manner. This includes phenotypic analyses of inactivated kinases for Mps3 NE localization, chromosome movement and telomere clustering, direct phosphorylation of the Mps3 luminal domain by these kinases in vitro, and normal NE localization in the phosphomimetic Mps3 mutant in the cdk/ddk background. The authors do not look at the phosphorylation of Mps3 in vivo in the inactivated kinase backgrounds. However, in the discussion, the authors discuss the likelihood that CDK and DDK are NOT responsible for phosphorylating Mps3 due to the luminal localization. This point needs to be clarified with a better explanation for the role of CDK and DDK. At a minimum, the authors should indicate on page 10 that "... although CDK/DDK can phosphorylate Mps3 in vitro they are unlikely to phosphorylate Mps3 in vivo (see discussion)."

We revised the text to distinguish the contribution of the phosphorylation in the luminal region of Mps3 from CDK/DDK-dependent phosphorylation in Mps3 localization. Particularly, the *mps3-AAA* mutant (deficient in the phosphorylation in the luminal region) shows defective NE localization with reduced focus-formation in NE, which is clearly different from resolution defects conferred by CDK/DDK inactivation. We

emphasize this point together with a revised model in a new model Figure (Figure 8).

Furthermore, to clarify the relationship of S189, S190 phosphorylation with CDK and/or DDK, we created new mutant alleles of the *mps3-S189A* (a putative DDK site mutant) and *mps3-S190A* (a putative CDK site mutant) mutants and characterized it in details (new Figure 6). As shown in the Figure, the *mps3-S189A* and *mps3-S190A* mutants did not show similar defects seen in CDK/DDK inactivation conditions (little resolution defects, rather showed localization defects); the *mps3-S189A* exhibited more severe in the defects than *mps3-S190A,* but both mutants showed more milder defects in Mps3 dynamics than the *mps3-AAA* mutant. Taken together, we suggest that phosphorylation at S189 and S190 are NOT mediated by CDK and DDK, respectively (CDK/DDK target(s) could be other proteins-See Discussion, page 20-21, line 638-654).

4. How much of the Mps3 protein is phosphorylated? This is relevant to the generation of the Mps3-DDD version and the finding that there are no mitotic defects. The authors should be able to resolve this point by quantitative western blotting in the different conditions. It is also unclear based on the provided images whether Mps3-DDD still localizes to the SPB – this should be addressed.

We have tried quantified how much S189 and S190 are simultaneously phosphorylated in total Mps3 protein in vivo by several methods including western blotting suggested by the reviewers, but could not get convincing results at this point (our anti-phospho-S189, S190 antibody works only in IP factions of Mps3-Flag and we do not have any positive control such as purified phosphorylated full-length Mps3 protein). After consulting with Mass spectrometry specialists, we found quantitative Mass spectrometry of heterogenous multiple phosphorylation is technically difficult, probably due to possible heterogenous phosphorylation status; single, double, triple in the TSS sequence as well as neighboring serine residues (8 T/S residues from 180-200 aa).

To support the idea on multiple different phosphorylated Mps3 molecules in an indirect way, as shown above, we studied the phenotypes of two new *mps3* point mutants in the phosphorylation sites in the lumen; *mps3-S189A* and *mps3-S190A* single mutants and found that the two mutants showed a defect in the localization (Figure 6), suggesting that single phosphorylation in the luminal region of Mps3 promotes NE localization.

As pointed out, Mps3-DDD does not show NE localization in mitotic cells, suggesting that DDD substitution is not enough to induce NE localization of Mps3, implying possible phosphorylation on the other sites. Alternatively, an additional layer of the regulation could be operating. In fact, we DO see mitotic NE localization of Mps3-DDD in *cdc7 bob1-1* (but not *cdc7-as3* with PP1; The localization in the cells, uniform NE staining, is different from focus/patch staining in meiotic cells; Figure 5G), suggesting the presence of negative regulation by DDK during vegetative growth. These points are discussed in the text (line 681-684, page 22). We believe complicated phospho-dependent regulation of Mps3 localization for its meiosis-specific NE localization, which is now emphasized in a new revised tile of the manuscript.

We hope that the reviewers can acknowledge our reasoning on what we are not able to quantify amounts of phosphorylated Mps3 in meiotic cells. Importantly, even without this kind data, we believe that our conclusion in the paper is still valid.

We showed that Mps3-DDD is localized to SPB in mitosis (double staining with tubulin), which is shown in Supplemental Figure S3A.

5. The biophysical data of Figure 6B are interpreted as showing that DDD peptides exhibit increased repulsion between aspartate side-chains of different peptides and between aspartate side-chains and acidic lipids (lines 475-477). Shouldn't an increase in repulsion between side-chains result in increased fluorescence in absence of PIP2, owing to a reduction in self-quenching? This isn't apparent in the data, in which DDD appears to have slightly lower fluorescence than WT at 0 uM. The second conclusion of increased repulsion between side-chains and acidic lipids appears reasonable and is supported by the MD data. This should be addressed by providing further data that support the first conclusion, or removing it and focussing solely on the second conclusion.

We agree with the suggestion. We excluded the first possibility of the repulsion among the DDD peptides and rewrote the text (line 525-534, page17). Moreover, as a support for MD simulation, we uploaded videos of MD analysis of Mps3-wild type, -DDD and -phosphorylated peptides (Videos 26-28).

6. The authors present a model of the interaction between SUN-KASH domains of the LINC complex based on in vitro liposome experiments in combination with molecular modelling. This is the novel part of the manuscript and if correct is an important finding. While the manuscript is focused on the role of phosphorylation in changing its dynamics, rather than the architecture of the LINC complex directly, figure 7 needs to be revised to show exactly what is known and what is speculation. Ideally, it would be desirable to substantiate the model with in vivo evidence, for example by super-resolution imaging. However, it is understood that the experiments are technically challenging as the NE is 20-nm wide so picking out this architecture by super-resolution microscopy would be a significant achievement. In addition, to make the mechanistic claim that phosphorylation is necessary for the interaction to occur (as is suggested in the figure) analysis of Mps2-Mps3 interaction in vivo in the presence of phosphorylation mutants would be required. The authors should clarify the caveats of their model in figure 7, either with appropriate changes to the text, or by the provision of additional data to support it.

Rather doing more experiments (Since Mps3-Mps2 complex is present in SPB in both meiotic and mitotic cells, IP using Mps3 phosphorylation-defective mutant protein during meiosis would not show drastic changes in the complex formation), we softened our conclusion by a revised model Figure (new Figure 8), we added a schematic drawing of the pathway to regulate Mps3 localization on NE at a single nucleus resolution (the upper part), which reflect our results more precisely in addition to the previous (truly putative) model shown in the bottom.

We also put a label of “?” in the case of our speculation. And, in the legend, we explained the model more in details by mentioning our caveats in the model.

7. The text needs extensive revision. It is frequently difficult to follow and contains numerous errors, mis-labeled figures, omitted error bars etc. – it should be carefully edited for grammar and to improve clarity and precision. The discussion is unusually long and includes much published information that is not necessary for contextualising the findings. It also needs substantial editing to maximise clarity and brevity. Some are listed below, but this is not an exhaustive list.

We rewrote the whole text and asked English editing service (Editage; https://www.editage.jp) to check the grammar and we also checked the manuscript more carefully such as mislabeling etc. Hope that this version will be much more readable than the previous version.